# CMMLU: Measuring massive multitask language understanding in Chinese

## Abstract

As the capabilities of large language models (LLMs) continue to advance, evaluating their performance is becoming simultaneously more important and more challenging. This paper aims to address this issue for Mandarin Chinese in the form of CMMLU, a comprehensive Chinese benchmark that covers various subjects, including natural sciences, social sciences, engineering, and the humanities. We conduct a thorough evaluation of more than 20 contemporary multilingual and Chinese LLMs, assessing their performance across different subjects and settings. The results reveal that most existing LLMs struggle to achieve an accuracy of 60% even, which is the pass mark for Chinese exams. This highlights that there is significant room for improvement in the capabilities of LLMs. Additionally, we conduct extensive experiments to identify factors impacting the models' performance and propose directions for enhancing LLMs. CMMLU fills the gap in evaluating the knowledge and reasoning capabilities of large language models in the Chinese context.

## 1 Introduction

Large language models (LLMs) have driven remarkable advancements in natural language processing and artificial intelligence, revolutionizing the field (Zhang et al., 2022; Scao et al., 2022; Zeng et al., 2023; Touvron et al., 2023a; OpenAI, 2023; Wu et al., 2023; Taori et al., 2023; Li et al., 2023a). However, assessing the knowledge and reasoning abilities of these models has become increasingly challenging, especially with the proliferation of LLMs that generate fluent and plausible responses.

To this end, researchers have created various benchmarks intended to evaluate different model capabilities (Wang et al., 2019b;a; Lin et al., 2022; Zellers et al., 2019; Hendrycks et al., 2021b; Chen et al., 2021). Specifically, Hendrycks et al. (2021a) proposed MMLU, a benchmark that encompasses various tasks ranging from elementary mathematics and computer science to management and law, which can be used to comprehensively measure LLM capabilities in terms of the knowledge embedded in them. Due to its multiple-choice question format, which facilitates easy evaluation, and the breadth of subject areas it encompasses, it has become widely used as a fundamental assessment tool of the knowledge encoded by LLMs. However, this benchmark is in English, which limits its ability to assess LLMs in other languages. Although some researchers (OpenAI, 2023) have attempted to automatically translate it to evaluate LLMs in other languages, the inherent bias towards Western (and specifically US) culture in the dataset renders it unsuitable and even inappropriate for assessing LLMs across diverse cultures and languages.

In this paper, we propose CMMLU (Figure 1), a comprehensive Chinese assessment suite specifically designed to evaluate the advanced knowledge and reasoning abilities of LLMs in a Chinese linguistic and cultural context. CMMLU covers a wide range of subjects, comprising 67 topics from elementary to advanced professional levels. It includes subjects that require computational expertise, such as physics and mathematics, as well as disciplines within the humanities and social sciences. Many of these tasks are not easily translatable from other languages due to their specific contextual nuances and wording. Furthermore, numerous tasks within CMMLU have answers specific to China, which may not be universally applicable or considered correct in other regions or languages.

We assess GPT4, ChatGPT, and more than 20 advanced open-source multilingual and Chinese LLMs on CMMLU. The results reveal that the majority of these models struggle to achieve an accuracy score of 60%, relative to random accuracy of 25%. Notably, GPT4 achieves an average accuracy of

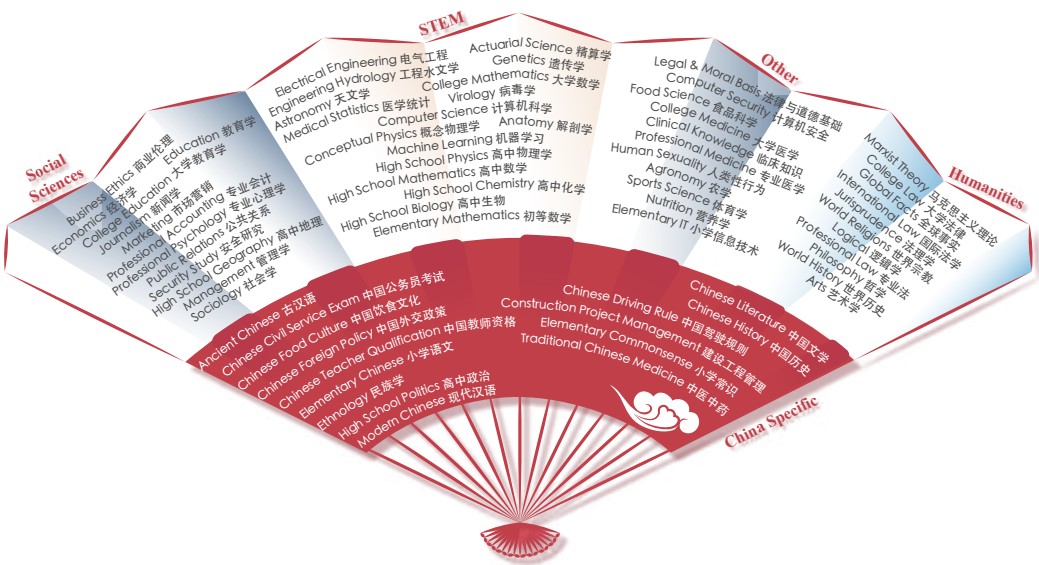

Figure 1: CMMLU task overview.

71%. These findings highlight the considerable room for improvement in LLMs in terms of Chinese knowledge and language understanding.

To gain a deeper understanding of the proficiency of the models in handling Chinese knowledge, we conduct a comprehensive analysis. We first focus on examining model performance across various subjects and find that all models exhibit uneven performance across different subjects, with comparatively higher scores in humanities and social sciences, but lower scores in China-specific and STEM subjects.

Furthermore, through extensive experiments, we find that: (1) most existing models do not benefit from chain-of-thought prompts in CMMLU; (2) few-shot examples help foundation models in the comprehension of tasks and enhance their reasoning abilities but do not help models that have undergone supervised fine-tuning (SFT) or reinforcement learning from human feedback (RLHF); (3) LLMs perform worse on questions with negation words compared to those without negation words, but recently-released models mitigate this disparity either through better pre-training data or fine-tuning; and (4) questions with sub-options (Section 4.2) are difficult for all existing LLMs, with even GPT4 dropping 20% in accuracy over such questions.

## 2 RELATED WORK

Benchmarking plays a crucial role in measuring AI development, particularly in the domain of LLMs. While benchmarks such as GLUE (Wang et al., 2019b) and SuperGLUE (Wang et al., 2019a) have played an important role in tracking progress in natural language understanding (NLU) tasks, they primarily focus on specific language skills. With an increasing move to generative models which are highly adept at generating fluent outputs, the value of these benchmarks has diminished, and new datasets have been proposed to evaluate LLM abilities over more general tasks, such as reading comprehension (Rajpurkar et al., 2018; Kwiatkowski et al., 2019; Li et al., 2022), summarization (Hermann et al., 2015), commonsense reasoning (Clark et al., 2018; Talmor et al., 2019; Sakaguchi et al., 2020), mathematical reasoning (Hendrycks et al., 2021b; Cobbe et al., 2021), and code generation (Chen et al., 2021; Austin et al., 2021).

In order to comprehensively assess the capabilities of LLMs, some benchmarks have incorporated massive multi-task evaluations into their frameworks (Hendrycks et al., 2021a; Liang et al., 2022; Srivastava et al., 2023). An example is MMLU (Hendrycks et al., 2021a), which includes multiple domains and tasks based on real-world exams. It has become very popular for LLM evaluation due to

its standardized and simplified format, comprehensive nature, and real-world relevance. However, all aforementioned benchmarks are primarily focused on English.

Given that Chinese is the language with the largest number of speakers worldwide, several benchmarks have been proposed for Chinese LLM evaluation. Following in the footsteps of GLUE and SuperGLUE, Xu et al. (2020) introduced CLUE, a benchmark for Chinese NLU that is widely used today. They also recently proposed SuperCLUE (Xu et al., 2023), which specifically focuses on LLMs. Recently, several Chinese benchmarks have emerged that follow the MMLU style, all of which are concurrent work with ours. In detail, Zhang & Li (2023) proposed ACLUE, focusing on ancient Chinese language understanding. Zeng (2023) presented MMCU, which covers four major domains (medicine, law, psychology, and education), with a particular focus on medicine and education. AGIEval (Zhong et al., 2023) provides problems from both Chinese and English standardized exams. C-Eval (Huang et al., 2023) and M3KE (Liu et al., 2023) collect more than 50 tasks from standard exams in China, while C-Eval covers various professions, and M3KE focuses on education examinations.

Compared to these benchmarks, CMMLU has several distinct features. Firstly, it includes more than 10 subjects that are not typically found in standard exams but are relevant to daily life, such as *Chinese food culture*, and *Chinese driving rules*. Secondly, it covers not only China-specific knowledge but also general world knowledge, such as *world religion*, *world history*, and *global facts*. Lastly, we have made our data completely public, enabling the community to evaluate their models freely and conveniently. A detailed comparison between CMMLU and other concurrent benchmarks is provided in Appendix A.

## 3 CMMLU

**Task Overview**   We created an extensive multitask test for Mandarin Chinese, which covers diverse areas of knowledge, including the humanities, social sciences, STEM (science, technology, engineering, and mathematics), and other areas that are important in daily life. It includes common test questions in subjects like mathematics, physics, and chemistry with answers that are not language or region specific, but also several tasks that are very region-specific, such as *Chinese driving rules*, *Chinese food culture*, and *Chinese teacher qualifications*. The questions in these tasks involve lots of China-related knowledge and can test a model's understanding and adaptability to Chinese. In addition, CMMLU also contains tasks that can only expressed in Chinese, such as *ancient Chinese language* and *Chinese literature*. The terms and concepts involved in these tasks heavily rely on Chinese expression and are almost impossible to be obtained from translation. The full list of subjects, the concepts tested in each subject, the number of questions, and the statistics of question and answer lengths are provided in Appendix B.

**Data collection**   We hired four annotators with undergraduate or higher education levels to manually collect the questions and answers from freely available resources, at a rate of 50 CNY per hour. To prevent our questions from appearing in the training set of LLMs, we invested specific effort in identifying non-publicly available materials, mock exam questions, and questions from quiz shows. More than 80% of our data was crawled from PDFs (after OCR), which further reduces the possibility of it occurring in LLM training data. The entire collection process took around 250 hours.

**Format**   Each question in the dataset is a multiple-choice question with 4 choices, only one of which is correct; see Figure 2 for an example. The questions are expressed as fill–in–the–blank (by choosing the correct option), or direct-answer questions. For chemical formulae and mathematical expressions, we use a 50:50 mixture of LaTeX and plain text, where plain text was only allowed if an expression is commonly used and not prone to ambiguity (as judged by the annotators). For instance, the chemical expression for water can be written in plain text as *H2O*, or in LaTeX format as $H_{2}O$.

**Quality Check**   To further check data quality, we randomly sampled 5% questions with answers for each subject, and conduct detailed verification through online resources. We estimate that there is around 2% of noise in the data, in terms of the correct answer not being present or being incorrectly labeled. Based on the results in Section 4 that most models struggle to achieve an average accuracy of 60%, we believe such an error rate does not compromise the overall results.

以下是关于 高中生物 的单项选择题，请直接给出正确答案的选项。

(Here are some single-choice questions about high school biology , please provide the correct answer choice directly.)

题目：同一物种的两类细胞各产生一种分泌蛋白，组成这两种蛋白质的各种氨基酸含量相同，但排列顺序不同。其原因是参与这两种蛋白质合成的:

(Question: Two types of cells within the same species each produce a secretion protein. The various amino acids that make up these two proteins have the same composition but differ in their arrangement. The reason for this difference in arrangement in the synthesis of these two proteins is:)

A. tRNA种类不同 (Different types of tRNA)
B. 同一密码子所决定的氨基酸不同 (Different amino acids determined by the same codon)
C. mRNA碱基序列不同 (Different mRNA base sequences)
D. 核糖体成分不同 (Different ribosome components)
答案是：C (Answer: C)

... [other examples]

题目：某种植物病毒V是通过稻飞虱吸食水稻汁液在水稻间传播的。稻田中青蛙数量的增加可减少该病毒在水稻间的传播。下列叙述正确的是:

(Question: A certain plant virus, V, is transmitted between rice plants through the feeding of rice planthoppers. An increase in the number of frogs in the rice field can reduce the spread of this virus among the rice plants. The correct statement among the options provided would be:)

A. 青蛙与稻飞虱是捕食关系 (Frogs and rice planthoppers have a predatory relationship)
B. 水稻和病毒V是互利共生关系 (Rice plants and virus V have a mutualistic symbiotic relationship)
C. 病毒V与青蛙是寄生关系 (Virus V and frogs have a parasitic relationship)
D. 水稻与青蛙是竞争关系 (Rice plants and frogs have a competitive relationship)
答案是： (Answer:)

Figure 2: Prompt with few-shot examples from CMMLU. English translations are provided in the bracket for better readability.

**Statistics** CMMLU contains 11,528 questions across 67 subjects. Each subject has at least 105 questions, which we split into a few-shot development set with 5 questions, and a test set with more than 100 questions. In terms of task types, CMMLU comprises 17 STEM tasks, 13 humanities tasks, 22 social science tasks, and 15 other tasks. Among these, 16 tasks are China-specific, which means they either do not exist in other countries or regions, or their answers may be different in other places. We provide an example for each subject type in Appendix C.

## 4 EXPERIMENTS

To provide an overview of existing LLMs on language understanding within the context of Chinese, we evaluate two commercial LLMs and more than 20 open-source LLMs in different sizes, language orients, and stages (i.e. either foundation model or SFT/RLHF model). We analyse their performance and investigate several factors that could affect the performance of LLMs.

**Setup** Our goal is to assess the LLMs performance on CMMLU, which contains multiple-choice questions with one correct answer for each question. There have been several strategies to perform multiple-choice question-answering task. In this paper, for commercial models which we cannot get the weights (i.e., GPT4 and ChatGPT), we input the question with all candidate choices, allowing the model to generate the output, and use a series of regular expressions (regex) to match the model's prediction. We call this *free generation* strategy. For open-source models, we follow Hendrycks et al. (2021a) to input the question and choices, and prompt the model by asking the answer key. Then we obtain the logits of the next predicted token, and compare the probability among the 4 tokens: 'A', 'B', 'C', and 'D' and select the token with the highest probability as the model's choice. We named this as *next token prediction* strategy. Besides these two strategies, there is another way which is to select the answer with the lowest perplexity when concatenated with the question.

We compared different strategies in Appendix G, and found that next token prediction is the most efficient way. Therefore, for the majority of the remaining paper, we report the results of the next token prediction. However, for some analysis in Section 4.2, we use the free generation strategy. The regex is designed based on the observation of ChatGPT and ChatGLM responses. The detail of regex and matching algorithm is provided in Appendix H.

**Prompt** We introduce each question with the phrase "以下是关于[主题]的单项选择题，请直接给出正确答案的选项 (Here are some multiple-choice questions about [subject], please provide the correct answer choice directly)", and evaluate models in both zero-shot and few-shot settings.

Table 1: Five-shot accuracy of models. We report macro average accuracy over subjects within each category. "Overall" = macro average score over all subjects. "State" indicates whether the model is pre-trained (Base) or Fine-tuned to follow instructions (Chat). '*' indicate there are both Base and Chat model released, we choose the one with better overall accuracy. The first block is multilingual- or English-oriented models, and the second block is Chinese-oriented models. To save space, we didn't present models with an overall score lower than 30.

| Model | State | STEM | Humanities | Social Science | Other | China-specific | Average |
|---|---|---|---|---|---|---|---|
| GPT4 | Chat | 65.23 | 72.11 | 72.06 | 74.79 | 66.12 | 70.95 |
| ChatGPT | Chat | 47.81 | 55.68 | 56.50 | 62.66 | 50.69 | 55.51 |
| LLaMA2-70B* | Base | 44.11 | 57.05 | 55.63 | 56.65 | 48.01 | 53.21 |
| Falcon-40B | Base | 33.33 | 43.46 | 44.28 | 44.75 | 39.46 | 41.45 |
| LLaMA-65B | Base | 34.47 | 40.24 | 41.55 | 42.88 | 37.00 | 39.80 |
| LLaMA2-13B* | Base | 33.04 | 39.73 | 38.45 | 42.54 | 35.67 | 38.24 |
| BLOOMZ-7B | Chat | 30.56 | 39.10 | 38.59 | 40.32 | 37.15 | 37.04 |
| LLaMA-30B | Base | 29.69 | 33.68 | 34.08 | 37.40 | 30.68 | 33.63 |
| LLaMA2-7B* | Base | 30.03 | 34.76 | 33.72 | 33.62 | 30.12 | 32.96 |
| ZH$_{\text{LLaMA}}$-13B | Chat | 27.12 | 33.18 | 34.87 | 35.10 | 32.97 | 32.63 |
| BX$_{\text{LLaMA}}$-13B | Chat | 27.50 | 32.47 | 32.33 | 35.77 | 31.64 | 31.90 |
| LLaMA-13B | Base | 29.21 | 30.96 | 31.74 | 33.07 | 30.86 | 31.24 |
| Baichuan2-13B* | Base | 48.36 | 67.44 | 66.40 | 65.94 | 63.48 | 61.92 |
| Baichuan-13B* | Base | 42.38 | 61.61 | 60.44 | 59.26 | 56.62 | 55.82 |
| InternLM-20B* | Chat | 42.70 | 60.51 | 58.00 | 57.62 | 54.72 | 54.52 |
| Xverse-13B* | Chat | 41.65 | 55.72 | 57.47 | 57.32 | 52.32 | 53.08 |
| InternLM-7B* | Base | 41.71 | 54.43 | 56.42 | 55.38 | 53.11 | 52.07 |
| ChatGLM2-6B | Chat | 42.65 | 50.88 | 51.22 | 50.72 | 48.66 | 48.87 |
| BatGPT-15B | Chat | 41.68 | 50.14 | 50.78 | 48.68 | 46.93 | 47.88 |
| Baichuan-7B* | Base | 35.25 | 48.07 | 47.88 | 46.61 | 44.14 | 44.43 |
| ChatGLM-6B | Chat | 32.35 | 39.22 | 39.65 | 38.62 | 37.70 | 37.48 |
| Random | – | 25.00 | 25.00 | 25.00 | 25.00 | 25.00 | 25.00 |

For zero-shot evaluation, we present a question with choices directly after the prompt. For few-shot evaluation, we provide up to 5 demonstration examples with answers before the question. The prompt concludes with the phrase "答案是：(Answer:)", as shown in the example in Figure 2. If the context exceeds the model's maximum length with few-shot examples, we dynamically remove the longest examples by counting sub-tokens.

**Models**    we assessed more than 20 models in different sizes from 12 model families. For commercial models, we evaluated ChatGPT and GPT4, which are two of the strongest LLMs.[1] For open-sourced models, we selected (1) English and multilingual-oriented models: BLOOM-7.1B (Scao et al., 2022), BLOOMZ-7.1B (Muennighoff et al., 2022), LLaMA-7B/13B/30B/65B (Touvron et al., 2023a), Bactrian-X-LLaMA (BX$_{\text{LLaMA}}$)-7B/13B (Li et al., 2023a), Falcon-7B/40B (Almazrouei et al., 2023), LLaMA2-7B/13B/70B (Touvron et al., 2023b), Chinese-LLaMA (ZH$_{\text{LLaMA}}$)-7B/13B (Cui et al., 2023); (2) Chinese-oriented models: Baichuan-7/13B and Baichuan2-7/13B (Yang et al., 2023), ChatGLM-6B and ChatGLM2-6B (Zeng et al., 2023), Xverse-13B,[2] InternLM-7B/20B (Team, 2023), MOSS-SFT-16B (OpenLMLab, 2023), Chinese-GLM-10B (Du et al., 2022), BatGPT-15B (Li et al., 2023b). The details about these models are provided in Appendix F.

## 4.1    MAIN RESULTS

Table 1 shows the performance of all models under the five-shot setting. Since the zero-shot results are similar to the five-shot results, we provide them in Appendix J.1.

**By model**    From the first block of the table, we observe the following: (1) LLaMA2-70B is the best open-sourced multilingual model, achieving an average accuracy of 53.21%, coming close to the

---

[1]The evaluation was conducted in May for ChatGPT and July for GPT4, 2023.

[2]https://github.com/xverse-ai/XVERSE-13B

ChatGPT performance at 55.51%. However, there is still a significant gap between LLaMA2-70B and GPT4 (70.95%); (2) 7B pre-trained multilingual models (except LLaMA2-7B) achieve nearly random results of 25% (since it's lower than 30%, they are not displayed in the table); (3) For those multilingual models, fine-tuning using Chinese resources consistently improves their performance (BX$_{LLaMA}$ and ZH$_{LLaMA}$ vs. LLaMA, BLOOMZ vs. BLOOM).

From the second block, we find that: (1) Among the Chinese LLMs, Baichuan2-13B demonstrates the best overall performance (beats ChatGPT) with only 13B parameters. We attribute it to the high quality of the training data; (2) Several Chinese LLMs achieve competitive results compared to LLaMA2-70B with less than 20B parameters. This demonstrates that when focusing on a single language, high-quality monolingual (or bilingual) training data can empower small models (7B or 13B) with good capability compared to multilingual training data. An overall observation is that models from the same family always improve as the model size increases.

**By subject** From the perspective of subject type, all models exhibit relatively high performance in humanities, social sciences, and other subjects, and medium performance in China-specific subjects, while low performance in STEM subjects. We attribute this to the nature of each subject type, and the capability of LLMs: (a) humanities, social sciences assess more on memorization which is relatively easy for LLMs; (b) China-specific topics encompass information that is either absent from the training data or inconsistent in multilingual training data; (c) STEM topics usually require complex reasoning, which has been proven to be difficult for existing LLMs. As expected, Chinese LLMs exhibit smaller gaps between China-specific subjects and other categories.

We compare the performance of the best-performing Chinese model, Baichuan2-13B, with the best-performing multilingual model, GPT4, for each subject. We categorize the subjects and present the results in Figure 3. The numerical results can be found in Appendix J.2.

From the figure, we note that the model's performance appears to be unbalanced, excelling in certain subjects but struggling in others. Specifically, *ancient Chinese* and *college actuarial science* are the most challenging subjects for both Baichuan2 and GPT4, yielding slightly better results than random, while the *legal and moral basis* is one of the easiest subjects for both models. When comparing the two models, we find that for most subjects, GPT4 outperforms Baichuan2 by a significant margin, while Baichuan2 surpasses GPT4 in 8 subjects, 6 of these are China-specific subjects, and the other 2 (*arts* and *philosophy*) contain a large amount of Chinese elements.[3] These findings suggest that including region- and culture-specific data in training is essential to accommodate users with different language backgrounds.

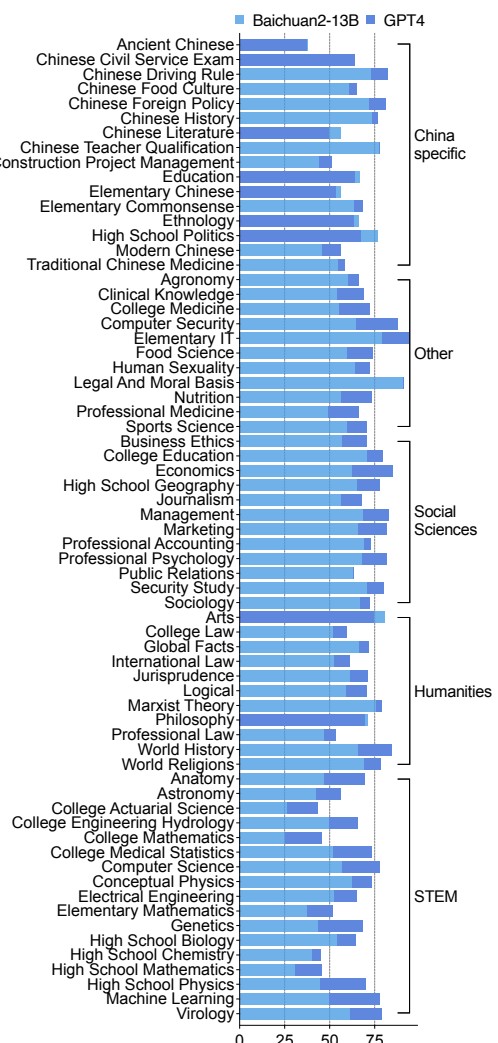

Figure 3: GPT4 vs. Baichuan2-13B-Chat on each subject (zero-shot). For a fair comparison, we use free generation strategy for both models.

---

[3] Due to these subjects contain a mixture of Chinese elements and global elements, we did not categorize them as China-specific.

Table 2: Zero-shot accuracy on CMMLU STEM subset, and full set, with direct answer (DA) prompt and chain-of-thought (COT) prompt. To ensure a fair comparison, we use the free generation strategy. "E changes" = the proportion (%) of instances cannot been matched after using COT − the proportion (%) of that with DA prompt.

| Model | STEM | | Overall | | E changes |
|---|---|---|---|---|---|
| | DA | COT | DA | COT | |
| ChatGPT | 45.22 | 46.58 | 53.14 | 52.73 | +0.55 |
| ChatGLM2-6B | 42.42 | 42.56 | 49.61 | 49.34 | -0.21 |
| Baichuan2-13B-Chat | 45.18 | 42.70 | 58.77 | 52.82 | +3.85 |
| BatGPT-15B-sirius | 38.13 | 34.66 | 45.26 | 42.87 | +1.35 |
| InternLM-Chat-20B | 42.09 | 32.31 | 53.52 | 43.29 | +3.87 |
| Xverse-13B-Chat | 40.13 | 30.53 | 52.96 | 39.27 | +19.77 |

## 4.2 ANALYSIS

In order to gain a comprehensive understanding of the LLM's performance on CMMLU, we explored three factors that may enhance the model's performance and two factors that could potentially diminish its performance. Specifically, we investigated whether the following factors can improve the model's performance: (1) utilizing chain-of-thought prompts, (2) increasing the number of input examples, and (3) employing larger-sized models within the same family. Conversely, we explored whether the following factors make the task more challenging for LLMs: (4) questions containing negation words, and (5) questions with sub-options within them. For different analyses, we choose different models in different stages according to the relevance and result availability.

**Can chain-of-thought prompt improve model performance?** To investigate the potential benefits of chain-of-thought (COT) prompt in generating better results, we modified the prompt from "请直接给出正确答案的选项 (please provide the correct answer choice directly)" to "逐步分析并选出正确答案 (Analyze step by step and select the correct answer)." Since our dataset does not contain answer analysis, we adopt zero-shot setting for this experiment. The results are presented in Table 2, the breakdown of all sub-categories is provided in Appendix J.3.

From the table, we see that for most models, the use of chain-of-thought prompt does not lead to improvement. ChatGPT and ChatGLM2 slightly gain improvement after using COT prompt for STEM subject, despite that the overall accuracy still decreases. We manually checked the outputs and found that models either fail to explicitly generate the answer option after the analysis (instead generating the content of the answer), or generate complex context to wrap the choice, which leads to the failure of regex match. An obvious case is Xverse, compare to the direct answer prompt, the use of COT prompt results in an increase of 19.77% responses that cannot be matched by our regex.

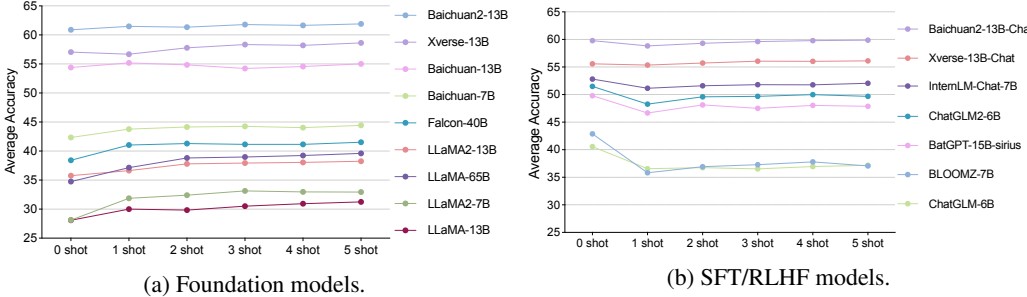

(a) Foundation models.  (b) SFT/RLHF models.

Figure 4: Overall accuracy of models with varying number of few-shot examples.

**Do few-shot examples help?** Many studies have shown that LLMs can benefit from the in-context examples, while some other studies have reported opposite observations (Liu et al., 2023; Zeng, 2023). In this context, we use CMMLU as a case study to investigate in-context learning (ICL) in LLM evaluation on multiple-choice questions.

As illustrated in Figure 4, we present the overall accuracy of models utilizing varying numbers of in-context examples. There is a clear discrepancy that, when provided with only one example, foundation models exhibit an overall boost, whereas fine-tuned models experience a decline in performance. We conjecture this is because foundation models are primarily optimized for natural text and may struggle to follow instructions. Providing examples helps these models better understand the task. In contrast, SFT/RLHF models are optimized to follow instructions, and the introduction of examples introduces a certain degree of mismatch with the data distribution during their fine-tuning, thus leading to a decline in performance.

When provided with more examples, while there may be fluctuations, the overall trend for foundation models indicates an improvement in performance with an increase in the number of examples. However, for fine-tuned models, there is no consistent trend.

**Impact of model size on performance**   We explored how the model's performance improves with an increase in the number of parameters. To this end, we examine several model families and present their five-shot accuracy in relation to model size in Figure 5.

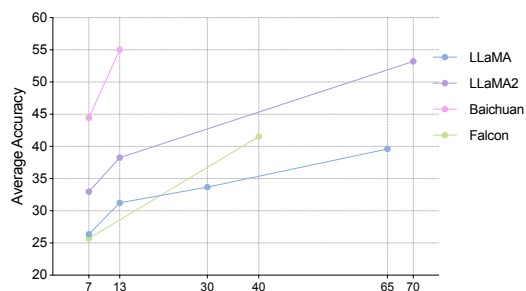

Figure 5: Five-shot accuracy of LLMs with different model sizes.

From the figure, we see that both LLaMA and LLaMA2 gain 5-point increase in scores as the model size changes from 7B to 13B, while Baichuan shows a remarkable 10-point improvement despite Baichuan-13B has 0.2T more training tokens than Baichuan-7B. We believe that have 7 billion parameters limit the model's capability in numerous tasks, while doubling the parameters to about 13 billion significantly enhances certain capabilities and improves memorization. As the model size continues to increase (as seen with LLaMA and LLaMA2), the efficiency of performance improvement decreases, with a 5x increase in model size resulting in a 7% improvement for LLaMA and a 15% improvement for LLaMA2. Comparing LLaMA2 and Baichuan, it becomes evident that a smaller model equipped with higher-quality monolingual training data not only can achieve but also surpass the performance of a larger model with insufficient monolingual training data in terms of monolingual performance.

Table 3: Average accuracy classified by questions w/ and w/o negation expressions, models are organized by model family. We use the free generation evaluation strategy.

| Model | 0-shot | | 5-shot | |
|---|---|---|---|---|
| | w/ | w/o | w/ | w/o |
| ChatGPT | 52.28 | 53.60 | 54.76 | 56.07 |
| GPT4 | 70.72 | 69.13 | 72.08 | 71.21 |
| LLaMA-65B | 22.94 | 36.54 | 37.09 | 40.18 |
| LLaMA2-13B | 24.16 | 37.27 | 30.32 | 39.49 |
| LLaMA2-13B-Chat | 28.24 | 37.90 | 34.40 | 38.73 |
| Baichuan-13B-Base | 47.84 | 55.47 | 51.20 | 56.03 |
| Baichuan2-13B-Base | 59.52 | 61.96 | 61.60 | 62.61 |
| Baichuan2-13B-Chat | 58.64 | 60.60 | 56.96 | 60.89 |
| ChatGLM-6B | 34.00 | 41.62 | 31.12 | 38.00 |
| ChatGLM2-6B | 51.20 | 51.88 | 50.08 | 50.04 |

Table 4: Average accuracy classified by questions w/ and w/o sub-options.   We use the free generation strategy, except for the models with "*", which are foundation models without instruction-following ability.

| Model | 0-shot | | 5-shot | |
|---|---|---|---|---|
| | w/ | w/o | w/ | w/o |
| GPT4 | 51.14 | 69.74 | 53.41 | 71.72 |
| ChatGPT | 34.85 | 53.90 | 33.33 | 56.47 |
| LLaMA2-70B* | 25.38 | 49.85 | 28.03 | 54.04 |
| Falcon-40B* | 23.11 | 38.72 | 28.41 | 42.14 |
| Baichuan2-13B-Chat | 47.73 | 59.78 | 34.09 | 57.41 |
| +COT | 35.61 | 54.61 | – | – |
| BatGPT-15B-sirius | 30.68 | 46.51 | 31.06 | 41.78 |
| +COT | 32.95 | 44.25 | – | – |
| ChatGLM2-6B | 28.79 | 50.84 | 27.65 | 49.82 |
| +COT | 36.74 | 50.18 | – | – |

**Are questions with negation more challenging?**   Previous research has pointed out that language models may encounter challenges with negation expression (Kassner & Schütze, 2020; Hosseini et al., 2021). To investigate whether this issue persists in the context of Chinese language and LLMs, we firstly employ string matching to classify the test set into questions with and without negation words. We then compare the performance of different models on these two subsets. Note that according to our string matching results, approximately 10.7% data contains negation expressions.

关于水平气压梯度力的说法正确的选项为：1 是形成风的直接原因；2 是大气作用在海平面上产生的压力；3 方向与等压线垂直；4 从高压指向低压
The correct option for the statement about the horizontal pressure gradient force is 1. It is the direct cause of the wind; 2. It is the pressure produced by the atmosphere on the sea level; 3. The direction is perpendicular to the isobar; 4. From high pressure to low pressure
A. 1234     B. 234     C. 134     D. 123
答案是：C (Answer: C)

Figure 6: An example of questions with sub-options. Example from high school geography.

In Table 3, we present 4 model families, from the table we find that most models (with the exception of GPT4 and ChatGLM2) perform less effectively on questions containing negative words compared to those without, aligning with the findings of previous studies, and highlights this common limitation of large language models.

Interestingly, developers have successfully mitigated this problem in different stages of development. For example, LLaMA2 demonstrates the enhancement of model's negation process ability using SFT/RLHF. The accuracy gap between question w/ and w/o negations decrease by about 5% after applying SFT/RLHF. Baichuan shows that better pre-training can also effectively alleviate this issue. Specifically, Baichuan2 reduces such a gap to 1-2% compared to Baichuan's 8-10% by using improved pre-training data. ChatGLM2 almost shows the same performance when answering questions with and without negations. We think the researcher has noticed the negation problem, and found that compared to complex reasoning ability, enhancing negative processing is relatively easy.

**Are questions with sub-options more challenging?** There is a typical question type in all kinds of Chinese exams called *sub-option questions*. These questions include a main statement along with multiple sub-options, and inquire about the count, order, or selection of the sub-options, which requiring the model to have deeper reasoning and inference skills (see example in Figure 6). The sub-options in CMMLU can appear in different formats, such as "a, b, c...; ①, ②, ③...", and account for about 10.8% of the dataset. We classified the data into two subsets based on sub-option presence, and put the evaluation results in Table 4. We observed that all these models performed weaker on sub-options questions compared to those without sub-options, with a decline ranging from 10% to 20%. Intuitively, the COT prompt should alleviate such a problem by guiding the model to analyze the sub-options one by one. However, the observation is that ChatGLM2 and BatGPT benefit from COT prompt while Baichuan doesn't.

## 5 CONCLUSION

We introduce CMMLU, a groundbreaking benchmark designed to assess the multi-task language understanding capabilities in Chinese. Our experimental findings reveal substantial opportunities for improvement within existing large language models. Through extensive analysis, we identify several factors that impact model performance and propose actionable directions for enhancing LLMs. We are confident that our benchmark dataset and analytical insights will empower researchers to effectively evaluate and design Chinese LLMs.

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

## A  COMPARISON TO CONCURRENT BENCHMARKS

C-Eval (Huang et al., 2023) and M3KE (Liu et al., 2023) are two similar benchmarks concurrent with our work. We compare the task distribution of these benchmarks in Table 5, and demonstrate that CMMLU contains more culture-related and region-related tasks. While there are differences in task distribution, we acknowledge that these datasets exhibit similarities in the task types and can, therefore, be jointly used as assessment criteria for evaluating the Chinese language capabilities of large models.

We further assess the overlap between CMMLU and both of these benchmarks. For this purpose, we first sort four choices for each question to eliminate the influence of choice order. Subsequently, we concatenate the question string with the sorted choice strings. Then, we remove all punctuation marks, including underscores and brackets, from the resulting strings. The final overlap, computed using exact string matching, yields a total of 74 for CEval and 158 for M3KE. This overlap accounts for approximately 1% of our dataset.

Table 5: Task distributions of contemporary similar datasets. CMMLU contains more subjects in humanities, social science, and others (usually country- or culture-specific) compared to CEval and M3KE, while fewer subjects in STEM. This indicates that our dataset is more inclined toward examining knowledge related to social, cultural, and regional factors.

| Model | STEM | Humanities | Social Science | Other | China-specific | Total |
|-------|------|------------|----------------|-------|----------------|-------|
| CEval | 20 | 11 | 10 | 11 | – | 52 |
| M3KE | **31** | 12 | 21 | 7 | – | **71** |
| CMMLU | 17 | **13** | **22** | **15** | **15** | 67 |

## B  CMMLU SUBJECTS

Table 6 lists all subjects of CMMLU. The table also provides details for each subject test, including the concepts covered, the supercategory to which each subject belongs, and the total number of questions.

Table 6: Summary of all 67 subjects. '*' indicate the China-specific subject. # Q means the total number of questions in this subject

| Task | Tested Concepts | Supercategory | # Q |
|---|---|---|---|
| Agronomy (农学) | Crop physiology, agroecology, soil science, breeding, ... | Other | 169 |
| Anatomy (解剖学) | Gross anatomy, neuroanatomy, clinical anatomy, ... | STEM | 148 |
| Ancient Chinese (古汉语)* | Classical Chinese, poems, words, songs,... | Social Science | 164 |
| Arts (艺术学) | Drama, poetry, ink painting, literature, movie, ... | Humanities | 160 |
| Astronomy (天文学) | Astronautics, planets, galaxies, asteroids, constellations, ... | STEM | 165 |
| Business Ethics (商业伦理) | Fairness and justice, transparency and accountability, ... | Social Science | 209 |
| Chinese History (中国历史)* | Ancient history, modern history, ancient culture, ... | Humanities | 323 |
| Chinese Literature (中国文学)* | Poetry, prose, drama, literary theory, ... | Humanities | 204 |
| Chinese Civil Service Exam (中国公务员考试)* | Science, law, Confucian classics, logic, common sense, ... | Social Science | 160 |
| Chinese Driving Rule (中国驾驶规则)* | Emergency procedures, signs, signals, traffic laws, ... | Other | 131 |
| Chinese Food Culture (中国饮食文化)* | Regional cuisines, cultural significance, nutrition, ... | Social Science | 136 |
| Chinese Foreign Policy (中国外交政策)* | China's foreign policy's principles, goals, history, ... | Social Science | 107 |
| Chinese Teacher Qualification (中国教师资格)* | Educational theory, pedagogy, psychology, language, ... | Social Science | 179 |
| Clinical Knowledge (临床知识) | Anatomy, physiology, healthcare, diagnose, pathology, ... | STEM | 237 |
| College Actuarial Science (大学精算学) | Factor reduction tables, density functions, ... | STEM | 106 |
| College Education (大学教育学) | Modern education, ancient education, school education, ... | Social Science | 107 |
| College Engineering Hydrology (大学工程水文学) | Air pressure, altitude, precipitation, ... | STEM | 106 |
| College Law (大学法律) | Criminal patterns, patent law, marriage law, ... | Humanities | 108 |
| College Mathematics (大学数学) | Matrices, derivatives, random variables, ... | STEM | 105 |
| College Medical Statistics (大学医学统计) | Probability, statistical tests, linear regression | STEM | 106 |
| College Medicine (大学医学) | Biochemistry, organic chemistry, genetics, metabolism, ... | STEM | 273 |
| Computer Science (计算机科学) | Data structures, algorithms, programming, operating systems, ... | STEM | 204 |
| Computer Security (计算机安全) | Network security, cryptography, firewalls, network protocols, ... | STEM | 171 |
| Conceptual Physics (概念物理学) | Mechanics, waves, power, energy, light, electricity, ... | STEM | 147 |
| Construction Project Management (建设工程管理)* | Planning, contracts, safety, budgeting, management, ... | Other | 139 |
| Economics (经济学) | Microeconomics, macroeconomics, economic systems, policy, ... | Social Science | 159 |
| Education (教育学) | Educational psychology, policies, technology, management ... | Social Science | 163 |
| Electrical Engineering (电气工程) | Electromagnetics, Ohm's Law, power Systems, ... | STEM | 172 |
| Elementary Chinese (小学语文)* | Ancient poems, classics, pronunciation, meaning, ... | Social Science | 252 |
| Elementary Commonsense (小学常识)* | heatstroke, fire, diet, first aid, ... | Other | 198 |
| Elementary Information and Technology (小学信息技术) | windows, word, powerpoint, ... | Other | 238 |
| Elementary Mathematics (初等数学) | Trigonometry, plane geometry, solid geometry, arithmetic, ... | STEM | 230 |
| Ethnology (民族学)* | Minority cultures, policies, religion, beliefs, history, ... | Social Science | 135 |
| Food Science (食品科学) | Chemistry, microbiology, processing, preservation, nutrition, ... | Other | 143 |
| Genetics (遗传学) | Mendelian Genetics, chromosomes, DNA, genetic disorders, ... | STEM | 176 |
| Global Facts (全球事实) | International economics, organizations, global events, ... | Humanities | 149 |
| High School Biology (高中生物) | Cell biology, genetics, evolution, ecology, microbiology, ... | STEM | 169 |
| High School Chemistry (高中化学) | Atomic, synthesis, chemical equilibrium, acid-base reactions, ... | STEM | 132 |
| High School Geography (高中地理) | Physical geography, human geography, environmental geography, ... | Social Science | 118 |
| High School Mathematics (高中数学) | Equations, trigonometry, analytic geometry, probability, ... | STEM | 164 |
| High School Physics (高中物理学) | Mechanics, heat, optics, electricity, acoustics, nuclear physics, ... | STEM | 110 |
| High School Politics (高中政治)* | Marxist philosophy, political economy, scientific socialism, ... | Social Science | 143 |
| Human Sexuality (人类性行为) | Reproductive health, contraceptive methods, mental health, ... | Other | 126 |
| International Law (国际法学) | Treaties, agreements, national sovereignty, law of the sea, ... | Humanities | 185 |
| Journalism (新闻学) | Media effects theory, communication models, journalism law, ... | Social Science | 172 |
| Jurisprudence (法理学) | Constitution, Administrative Law, Civil Law, Criminal Law, ... | Humanities | 411 |
| Legal And Moral Basis (法律与道德基础) | Legal ethics, moral views and values, social ethics, history, ... | Other | 214 |
| Logical (逻辑学) | Propositional logic, inductive reasoning, critical thinking, ... | Humanities | 123 |
| Machine Learning (机器学习) | Supervised learning, unsupervised learning, neural networks, ... | STEM | 122 |
| Management (管理学) | Organizational theory, leadership, international management, ... | Social Science | 210 |
| Marketing (市场营销) | Marketing Concepts, Pricing Strategies, Consumer Behavior, ... | Social Science | 180 |
| Marxist Theory (马克思主义理论) | Basic principles, Practical significance, contemporary value, ... | Humanities | 189 |
| Modern Chinese (现代汉语)* | Grammar, semantic, literature, ... | Social Science | 116 |
| Nutrition (营养学) | Dietary fiber, trace elements, fatty acids, ... | STEM | 145 |
| Philosophy (哲学) | Chinese Philosophy, Western Philosophy, Book of Changes, ... | Humanities | 105 |
| Professional Accounting (专业会计) | Audit, financing, assets, profit distribution, ... | Social Science | 175 |
| Professional Law (专业法学) | Patent Law, Criminal Law, Contract Law, ... | Humanities | 211 |
| Professional Medicine (专业医学) | Clinical Trials, Fractures, HIV, ... | STEM | 376 |
| Professional Psychology (专业心理学) | emotions, thought patterns, perception, ... | Social Science | 232 |
| Public Relations (公共关系) | Negotiations, Organizational Image, Etiquette, ... | Social Science | 174 |
| Security Study (安全研究) | national security, terrorism, ... | Social Science | 135 |
| Sociology (社会学) | Socialization, cities and community, ... | Social Science | 226 |
| Sports Science (体育学) | swimming, Chinese martial arts, heart rate, ... | Other | 165 |
| Traditional Chinese Medicine (中医中药)* | human meridians, yin and yang, ... | Other | 185 |
| Virology (病毒学) | Pathogen, viral gene mutation, infection | STEM | 169 |
| World History (世界历史) | Ancient civilizations, the Industrial Revolution, world wars, ... | Humanities | 161 |
| World Religions (世界宗教) | Islam, Judaism, Buddhism, Christianity, ... | Humanities | 160 |

Table 7 presents the breakdown of statistical results of the CMMLU test set for each supercategory, including the number of tasks, number of questions, average question counts for each subject, maximum and minimum counts of questions, and average token length for question and choices. Meanwhile, Figure 7 provides a visualization of the token lengths of questions and answers for each subject.

Table 7: The statistics of the CMMLU test set, where Q represents the question and C indicates the answer choices.

| Subject | Tasks | #Q | Avg. #Q | Max. #Q | Min.#Q | Avg.Q Tokens | Avg.C Tokens |
|---|---|---|---|---|---|---|---|
| STEM | 17 | 2531 | 148.88 | 230 | 105 | 38.53 | 11.62 |
| Humanities | 13 | 2489 | 191.46 | 411 | 105 | 41.65 | 10.10 |
| Social Science | 22 | 3652 | 166.00 | 252 | 107 | 36.84 | 7.25 |
| Other | 15 | 2910 | 194.00 | 376 | 126 | 31.31 | 7.02 |
| China-specific | 15 | 2572 | 171.46 | 323 | 107 | 44.54 | 8.20 |
| All | 67 | 11582 | 172.87 | 411 | 105 | 36.85 | 8.76 |

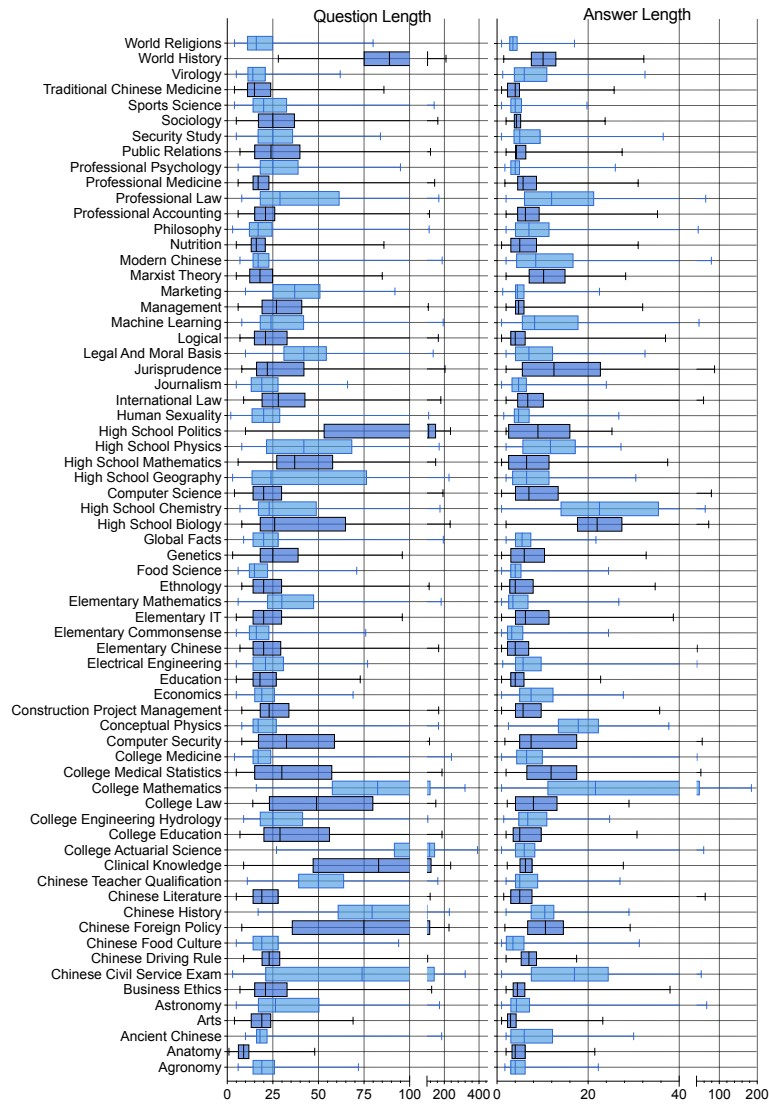

Figure 7: Question and answer lengths of each subject.

## C    CMMLU EXAMPLES

Table 8: Examples with their corresponding English translations from CMMLU among different subjects, where the bold items indicate the correct choices.

| Subject | Question | Choices |
|---|---|---|
| STEM | 油罐车后面都有一条拖地的铁链，其作用是？ | A. 作为油罐车的标志
B. 向外界散热
C. 发出响声，提示其他车辆和行人
**D. 把电荷导入大地，避免由静电造成的危害** |
| | What is the purpose of the iron chain dragging on the ground behind an oil tanker? | A. As a symbol of an oil tanker
B. Dissipating heat to the outside world
C. Emitting sound to alert other vehicles and pedestrians
**D. Conducting electric charges into the ground to prevent hazards caused by static electricity** |
| Humanities | 长篇小说《京华烟云》的作者是？ | A. 丁玲
B. 柔石
**C. 林语堂**
D. 老舍 |
| | Who is the author of the novel "Moment in Peking"? | A. Ding Ling
B. Rou Shi
**C. Lin Yutang**
D. Lao She |
| Social Science | "抓饭"是（）的特色饮食 | A. 藏族
**B. 维吾尔族**
C. 苗族
D. 朝鲜族 |
| | "Pilaf" is a characteristic cuisine of () | A. Zang nationality
**B. Uygur**
C. Miao nationality
D. Chaoxian nationality |
| Other | 全身黄染是食用（）过量 | **A. 维生素A**
B. 维生素D
C. 维生素B
D. 维生素C |
| | The yellowing of the whole body is a result of excessive consumption of () | **A. Vitamin A**
B. Vitamin D
C. Vitamin B
D. Vitamin C |
| China specific | 孔子弟子中擅长做生意的是谁？ | **A. 子贡**
B. 子路
C. 颜回
D. 子张 |
| | Who among Confucius's disciples was good at doing business? | **A. Zi Gong**
B. Zi Lu
C. Yan Hui
D. Zi Zhang |

Table 8 provides examples from CMMLU in each category.

# D    CMMLU DIFFICULTY DISTRIBUTION

We analyze the difficulty distribution of CMMLU from two perspectives. Firstly, the CMMLU benchmark encompasses a diverse range of difficulty levels: 5 subjects at primary school level, 10 at middle/high school level, 23 at college level, and 29 at professional level, ensuring a comprehensive difficulty spectrum.

Secondly, to estimate the difficulty distribution within each subject, we evaluated the top 20 models from our main results table. Each question was treated as a data point, and we recorded the number of models correctly answering each question. This approach allowed us to map out the difficulty distribution across subjects.

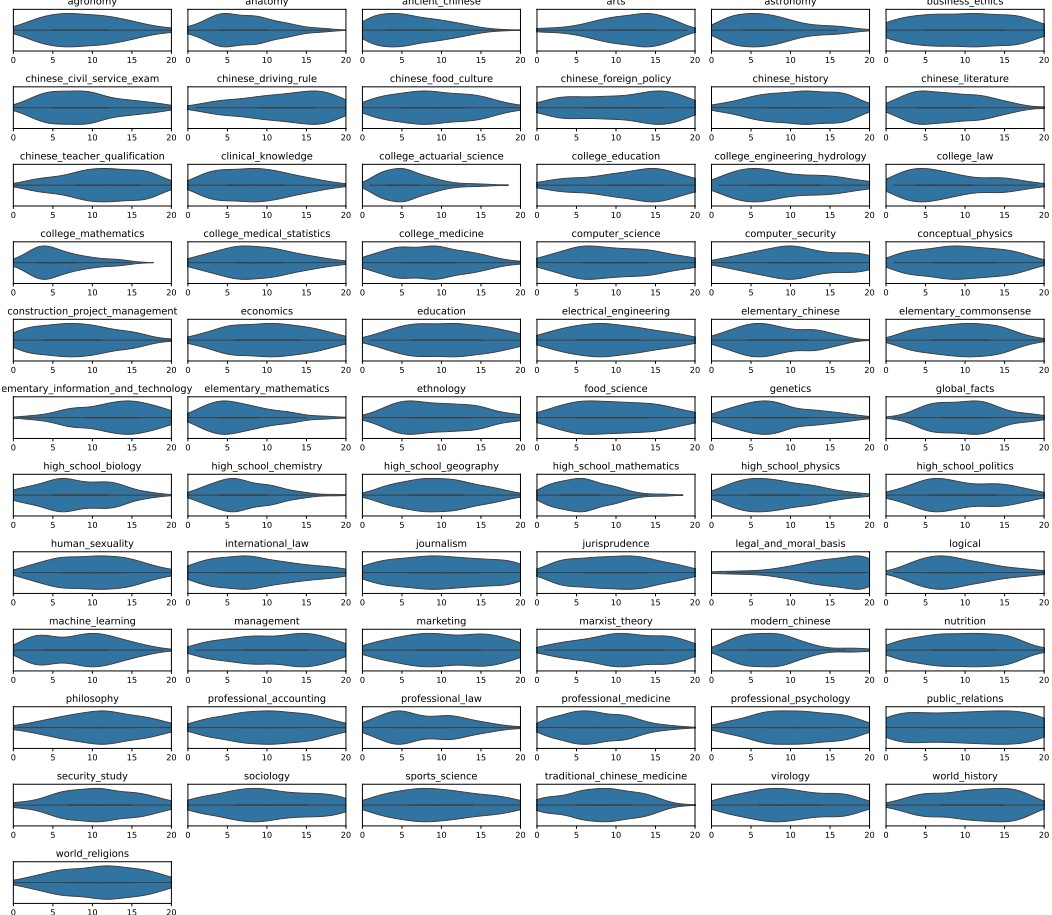

Figure 8: Difficulty distribution estimation of each subject. We use violin plot for visualization, where the x-axis represents the number of models that correctly answer a question, and the y-axis indicates the quantity of such questions. A peak on the left side of the plot (e.g., *college actuarial science* at position $[3, 3]$) suggests that the subject is generally challenging, as most questions are correctly answered by only a few models. Conversely, a peak on the right (e.g., *arts* at position $[1, 4]$) indicates a relatively simpler subject, where most questions are correctly answered by many models. Subjects exhibiting multi-peak distributions reveal a varied difficulty range within that subset. For instance, a hypothetical scenario with a dataset comprising basic arithmetic problems and complex calculus questions would result in a distribution with two distinct peaks separated by a notable gap, resembling a horizontal funnel. This indicates a wide spectrum of difficulty levels, from very easy to highly challenging.

Figure 8 reveals that the majority of subjects exhibit a single peak in their difficulty distribution. This single-peak pattern indicates a uniform level of difficulty within these subjects, suggesting a consistent challenge for models across the range of questions. However, certain subjects, such as *machine learning* (located at position $[9, 1]$) and *professional law* (at position $[10, 3]$), display dual

peaks. This dual-peak pattern signifies a notable presence of both relatively easy and challenging questions, with fewer intermediate-level questions. Despite the presence of two peaks, the transition between these peaks is gradual rather than abrupt, indicating a smooth progression in difficulty levels within these subjects.

# E EMERGENT ABILITY SHOWN IN CMMLU SUBJECTS

Figure 9: LLaMA-2 models performance on each subject. s, m, l means 7B, 13B and 70B models, respectively.

We assessed the concept of emergent ability using the LLaMA-2 model family. Figure 9 illustrates the performance of the LLaMA-2 pre-trained models (7B, 13B, and 70B) across various subjects. The figure indicates that, for most subjects, there is a correlation between increased model size and enhanced performance. Notably, in subjects like *college education* (position $[2, 4]$), *elementary commonsense* (position $[3, 6]$), *human sexuality* (position $[4, 7]$), and *public relations* (position $[5, 12]$), the performance of the 7B and 13B models is comparable, while the 70B model shows a significant improvement.

However, since LLaMA-2-70B model has been trained on a more extensive dataset compared to its 7B and 13B counterparts, which likely includes more comprehensive coverage in these specific domains. We cannot simply attribute it to emergent ability. In addition, these tasks are mostly belongs to social science rather than STEM (which might need intensive reasoning). Given these complexities, we leave the exploration of emergent ability in our future research endeavors.

## F    MODELS BEING EVALUATED

**ChatGPT/GPT4**   are GPT models developed by OpenAI and fine-tuned using reinforcement learning from human feedback (RLHF). As commercial products, specific details about the model size, training data, and training process remain undisclosed.

**Falcon**   is a decoder-only model created by TII and trained on 1,000B tokens of RefinedWeb (Penedo et al., 2023) data. Due to the high quality of its training data, Falcon-40B performs competitively with LLaMA-65B on various benchmarks.

**LLaMA**   is an auto-regressive language model proposed by Meta. It incorporates several structural improvements over the vanilla transformer and is trained on a mixture of publicly available data sources. LLaMA has demonstrated performance that is comparable to or even superior to models that are ten times its size.

**LLaMA2**   is an upgraded version of LLaMA developed by Meta. The preprocessing stage involves more robust data cleaning and updating data mixes, and the model employs a 40% increase in the total token count during training. Additionally, it up-samples the most factual sources to enhance knowledge and reduce hallucinations. Grouped-query attention (GQA) has been employed to reduce GPU memory usage.

**BLOOM**   is a multi-lingual targeted LLM developed by BigScience. It is trained on 46 natural languages and 13 programming languages. The largest BLOOM model consists of 176B parameters, but deploying such a large model can be challenging. In this paper, we evaluate the performance of the 7B BLOOM model.

**BLOOMZ**   is derived from BLOOM through fine-tuning on a cross-lingual task mixture (xP3), which is an instruction-following dataset. BLOOMZ exhibits competitive performance with models that have a larger number of parameters across various non-generation tasks.

**Bactrian-X**   is a series of LLMs (LLaMA, BLOOM, mT5) proposed by MBZUAI. These models are fine-tuned on a multilingual instruction-following dataset that encompasses 52 languages. All the fine-tuned Bactrian-X models demonstrate performance improvements compared to their corresponding base models in multilingual generation settings.

**ChatGLM and ChatGLM2**   are bidirectional dense models pre-trained using the General Language Model (GLM) algorithm developed by Tsinghua University. They support bilingual (Chinese and English) language processing. ChatGLM is a version of GLM that is enhanced with supervised fine-tuning, feedback bootstrap, and reinforcement learning with human feedback, specifically optimized for Chinese question answering (QA) and dialogue tasks. In this paper, we evaluate the performance of 10B and 6B models of GLM.

**BatGPT**   jointly developed by Wuhan University and Shanghai Jiaotong University, is a bilingual (Chinese and English) and bidirectional language model. BatGPT is initialized with a novel parameter expansion method, which enables it to absorb knowledge from the pre-training of other LLMs. With a bidirectional autoregressive architecture and further enhancement through Supervised Fine-Tuning (SFT) and Reinforcement Learning from Human and AI Feedback (RLHAF), BatGPT is able to handle long-range, multi-turn question-answering tasks effectively and alleviate concerns regarding memory limitations. The evaluation of the 15B version is presented in this work.

**MOSS-SFT**   is an open-source Chinese language model proposed by Fudan University. It is comparable to ChatGPT in terms of training scale and alignment techniques. MOSS-SFT is initialized with CodeGen and further pre-trained on 100B Chinese tokens and 20B English tokens. The Supervised Fine-Tuned (SFT) version of MOSS-SFT enables the model to follow instructions in multi-turn dialogues.

**Chinese-LLaMA** is part of the Chinese-LLaMA-Alpaca project, an open-source initiative that extends the vocabulary of LLaMA and Alpaca to include more Chinese tokens. The models are then further trained on a larger Chinese corpus to enhance their performance.

**Baichuan and Baichuan2** are large language model families publicly released by Baichuan Intelligent Technology. Both include versions with 7B and 13B parameters, as well as base and chat variants. Baichuan models are trained on high-quality corpora totaling 1.4 trillion tokens, which surpasses LLaMA-13B by 40%. The models offer support for both Chinese and English languages, and have an extensive context window of 4096. Baichuan2 series is trained on nearly twice the amount of high-quality data, resulting in additional performance enhancements.

**Xverse** is a 13B multilingual large language model developed by Shenzhen Yuanxiang Technology. It is trained on 1.4 trillion tokens from diverse sources and supports an extensive 8k context length, efficient tokenization, and advanced training technologies, making it both versatile and efficient.

**InternLM** is an open-source, lightweight training framework developed collaboratively by Shanghai AI Laboratory in partnership with researchers from various universities and companies. Its primary objective is to facilitate model pre-training without the need for extensive dependencies. Utilizing a unified codebase, it supports both large-scale cluster pre-training on thousands of GPUs and fine-tuning on a single GPU, achieving remarkable performance enhancements. Notably, InternLM achieves nearly 90% acceleration efficiency when training on 1024 GPUs. Based on the InternLM framework, a model family including 7B and 20B versions as well as base and chat variants was released.

## G STRATEGIES FOR ESTIMATING MODEL CHOICES

In this section, we compare three strategies for multiple-choice question evaluation. We introduce the mechanism of each strategy, explain its rationale, and compare their efficiency, strengths, and weaknesses. For convenience, we assume the question is "textQ", and the four choices are: "textA", "textB", "textC", "textD".

**Strategy 1 – Next Token Prediction** The idea is to input the question along with all candidate choices and prompt the model with a direct answer text, such as "The answer is: ". We then retrieve the probabilities of the next predicted token and compare these probabilities over the four choice indicator tokens, typically $[A, B, C, D]$. The token with the highest probability is treated as the model's choice.

- Example input:
    - *Question: textQ*
      *A. textA*
      *B. textB*
      *C. textC*
      *D. textD*
      *Answer:*
- Efficiency: High
- Pro: Most efficient method.
- Con: The model may not tend to generate a token from these choice letters.
- How to mitigate the cons: Provide few-shot examples with their expected answers.
- Works or frameworks use this strategy: MMLU (Hendrycks et al., 2021a), HELM (Liang et al., 2022).

**Strategy 2 – Perplexity Comparison** After combining question with all candidate choices. We concatenate each candidate answer with the full question and candidates text. These concatenated texts are then input to the model for a forward pass, and we compute the perplexity for each. The sequence with the lowest perplexity is treated as the model's choice.

- Example input (4 inputs):

  - *Question: textQ*
    *A. textA*
    *B. textB*
    *C. textC*
    *D. textD*
    *Answer: A. textA*
  - *Question: textQ*
    *A. textA*
    *B. textB*
    *C. textC*
    *D. textD*
    *Answer: B. textB*
  - *Question: textQ*
    *A. textA*
    *B. textB*
    *C. textC*
    *D. textD*
    *Answer: C. textC*
  - *Question: textQ*
    *A. textA*
    *B. textB*
    *C. textC*
    *D. textD*
    *Answer: D. textD*

- Efficiency: Low

- Pro: Aligns with the objective of language model optimization as perplexity reflects the true probability of a model generating the given text.

- Con: Low efficiency. Usually take 4x time (for a 4-choice question) compared to Next Token Prediction.

- How to mitigate the cons: Efficient implementation that only computes the same prefix once.

- Works or frameworks use this strategy: LM-Evaluation-Harness (Gao et al., 2021), Open-Compass.[4]

**Strategy 3 – Free Generation**  We input the question and candidate choices to the model and prompt it by asking for the correct choices. We allow the model to continue generating text, and then use the auxiliary method to match the patterns and extract the model's choices.

- Example input:

  - *Question: textQ*
    *A:textA*
    *B:textB*
    *C:textC*
    *D:textD*
    *Answer:*

- Efficiency: Medium/Low

- Pro: Allow various prompting,

- Con: Need answer extraction via human/model/regular expression. This process can be costly and error-prone. The generation can be very long, resulting in significant time consumption.

- How to mitigate the cons: Train a robust answer extraction model, or design robust regular expressions. Use a small temperature when doing generation.

---

[4]https://github.com/open-compass/opencompass

Table 9: Comparison of different evaluation strategies. We compare next token prediction (i.e. "Next"), and free generation ("Gen"). We also list the proportion of responses that cannot matched by our regex (% E). Note that our regex is designed based on the observation of ChatGPT and ChatGLM responses.

| Model | Next | Gen | % E |
|---|---|---|---|
| *0-shot* | | | |
| Baichuan2-13B-Chat | 59.79 | 58.77 | 0.71 |
| BatGPT-15B-sirius | 49.81 | 45.26 | 2.35 |
| ChatGLM-6B | 40.56 | 40.43 | 1.15 |
| ChatGLM2-6B | 51.48 | 49.61 | 1.51 |
| InternLM-Chat-20B | 55.06 | 53.52 | 0.01 |
| Xverse-13B-Chat | 55.59 | 52.96 | 0.88 |
| *5-shot* | | | |
| Baichuan2-13B-Chat | 59.89 | 54.44 | 6.44 |
| BatGPT-15B-sirius | 47.88 | 40.13 | 4.58 |
| ChatGLM-6B | 37.17 | 36.83 | 1.65 |
| ChatGLM2-6B | 49.69 | 48.80 | 0.56 |
| InternLM-Chat-20B | 54.52 | 51.51 | 0.42 |
| Xverse-13B-Chat | 56.12 | 51.64 | 5.55 |

- Works or frameworks use this strategy: OpenCompass, C-Eval (Huang et al., 2023).

Table 9 compares models performance using strategy 1 and strategy 3. Since strategy 2 is time-consuming, we didn't conduct results on it. From the table, we find that using next token prediction achieves a higher score than using the free generation strategy for all models, but the gap is less than 3% for most of the models under the zero-shot setting (with the exception of BatGPT which is about 5%). For both zero-shot and five-shot settings, the gap between strategy 1 and 2 is positively correlated to the proportion of the instances that cannot match any choice using regex. Hence, we believe using the next token prediction to force the model to make a choice among the given choices can effectively reflect its knowledge capacity.

## H    REGULAR EXPRESSIONS MATCHING ALGORITHMSL

The pseudocode in Algorithm 1 outlines the ExtractChoice function for extracting choices from an LLM output string.

Initially, the function examines whether the first character of the string corresponds to a valid choice and returns that choice if true. To accommodate the complex responses of different LL.M.s, we adopt a four-step matching mechanism.

First: Identify and extract choices by seeking patterns of some choice statements, such as the term "answer" (answer) followed by valid options. Second: Employ a pattern to recursively identify and extract the choices mentioned in the string, iterating until they finally appear. Third: Use weak single matching patterns. Fourth: Check for responses that mention a single choice.

If there is no matching pattern or unique selection, "E" is returned by default, indicating that no selection was confidently extracted.

**Algorithm 1** Algorithm for Extracting Choices from Response Strings

```
 1: procedure EXTRACTCHOICE(response)
 2:     response ← convert to string(response)
 3:     choices ← ['A','B','C','D']
 4:     if first character of response ∈ choices then
 5:         return first character of response
 6:     end if
 7:     patterns₁ ← [
 8:     (r'答案(选项)?(是|为)：? ?([ABCD])', 3),
 9:     (r'答案(是|为)选项 ?([ABCD])', 2),
10:     (r'故?选择?：? ?([ABCD])', 1),
11:     (r'([ABCD]) ?选?项(是|为)?正确', 1),
12:     (r'正确的?选项(是|为) ?([ABCD])', 2),
13:     (r'答案(应该)?(是|为)([ABCD])', 3),
14:     (r'选项 ?([ABCD]) ?(是|为)?正确', 1),
15:     (r'选择答案 ?([ABCD])', 1),
16:     (r'答案?：?([ABCD])', 1),
17:     (r'([ABCD])(选?项)?是?符合题意', 1),
18:     (r'答案选项：? ?([ABCD])', 1),
19:     (r'答案(选项)?应?该?为(.*?)([ABCD])', 3),
20:     (r'textbf{\(([ABCD])', 1)
21:     ]
22:     patterns₂ ← [
23:     (r'([ABCD])(.*?)当选', 1),
24:     (r'([ABCD])(.*?)正确', 1)
25:     ]
26:
27:     patterns₃ ← [
28:     (r'[^不]是：? ?([ABCD])', 1),
29:     (r'^选项([ABCD])', 1)
30:     ]
31:
32:     for each patterns in [patterns₁, patterns₂, patterns₃] do
33:         for each (pattern, idx) in patterns do
34:             if pattern is found in response then
35:                 answer ← matched group(idx)
36:                 if answer ∈ choices then
37:                     return answer
38:                 end if
39:             end if
40:         end for
41:     end for
42:     pattern₄ ← r'^[^ABCD]*([ABCD])[^ABCD]*$'
43:     if pattern₄ is matched in response then
44:         answer ← matched group(1)
45:         if answer ∈ choices then
46:             return answer
47:         end if
48:     end if
49:     return "E"                          ▷ Return E as default if no match is found
50: end procedure
```

# I CORRELATION TO OTHER BENCHMARKS

To investigate the correlation between models performance on CMMLU and other benchmarks, we choose 6 popular English LLMs and 5 benchmarks to conducte correlation analysis.

From Figure 10 we find that CMMLU demonstrates a strong correlation with four of these benchmarks, which span areas such as mathematics, commonsense reasoning, and coding. The exception is the PIQA task, where the relevance is somewhat diminished due to most models achieving high scores (>80%) on this task. However, 0.88 still shows strong positive correlation.

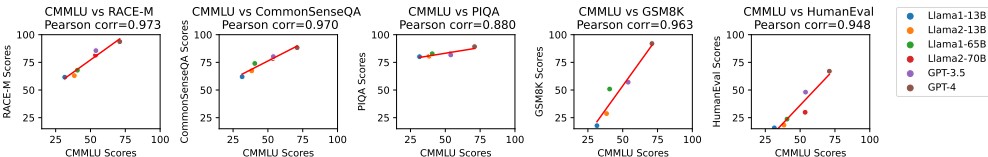

Figure 10: Correlation between the performance on CMMLU and that of other benchmarks. We choose RACE dataset for general language understanding, CommonSenseQA for commonsense reasoning,, PIQA for general reasoning, GSM8K for mathematics, and HumanEval for code ability.

# J BREAKDOWN OF MODEL PERFORMANCE

## J.1 RESULTS OF ZERO-SHOT

Table 11 displays zero-shot results of the LLMs on CMMLU by 5 sub-categories.

## J.2 THE RESULTS OF EACH SUBJECTS

We compared the 0-shot and 5-shot results of selected LLMs that showed higher performance on each subject in Table 10. We further analyze the performance distribution of multiple LLMs across all subjects in Figure 11. It is evident from the figure that LLMs with higher performance exhibit diverse abilities across various tasks, while those with lower performance face challenges in most subjects. Furthermore, the scatter plot distribution indicates comparable performance levels among LLMs across different subjects.

## J.3 THE EFFECT OF CHAIN-OF-THOUGHT PROMPT

Table 12 shows the breakdown of the models performance after using chain-of-thought prompt.

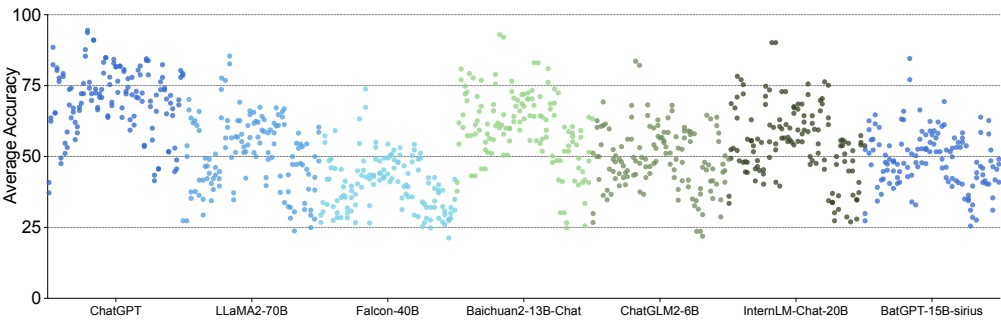

Figure 11: The performance of selected LLMs on CMMLU on each subject. The results for both 0-shot and 5-shot scenarios are depicted.

Table 10: The results of 0-shot and 5-shot accuracy per subject. The number on the left of 0-shot and the number on the right of 5-shot. The models are LLaMA2-70B, Falcon-40B, Baichuan2-13B-Chat, ChatGLM2-6B, InternLM-Chat-20B, BatGPT-15B-sirius.

| Subject | GPT4 | LLaMA2 | Falcon | Baichuan2 | ChatGLM2 | InternLM | BatGPT |
|---|---|---|---|---|---|---|---|
| Ancient Chinese | 37.2 / 40.9 | 27.4 / 27.4 | 26.8 / 29.3 | 40.9 / 37.8 | 26.8 / 29.9 | 33.5 / 36.0 | 29.9 / 27.4 |
| Chinese Civil Service Exam | 63.7 / 62.5 | 50.0 / 53.8 | 33.8 / 30.6 | 61.9 / 54.4 | 51.2 / 50.0 | 49.4 / 52.5 | 52.5 / 51.2 |
| Chinese Driving Rule | 82.4 / 88.5 | 66.4 / 70.2 | 55.0 / 57.3 | 77.1 / 80.9 | 60.3 / 62.6 | 67.2 / 68.7 | 62.6 / 59.5 |
| Chinese Food Culture | 65.4 / 65.4 | 35.3 / 37.5 | 33.1 / 41.9 | 60.3 / 64.7 | 50.0 / 41.9 | 52.2 / 52.9 | 55.9 / 47.1 |
| Chinese Foreign Policy | 81.3 / 80.4 | 62.6 / 63.6 | 48.6 / 42.1 | 74.8 / 72.0 | 60.7 / 54.2 | 71.0 / 63.6 | 52.3 / 56.1 |
| Chinese History | 76.5 / 77.7 | 61.9 / 61.0 | 46.1 / 49.2 | 72.8 / 69.7 | 61.0 / 69.3 | 77.1 / 78.3 | 61.6 / 64.7 |
| Chinese Literature | 49.5 / 47.5 | 37.7 / 36.3 | 27.5 / 32.4 | 57.4 / 57.4 | 36.3 / 34.8 | 48.0 / 48.5 | 39.2 / 34.3 |
| Chinese Teacher Qualification | 78.2 / 79.3 | 59.2 / 65.9 | 45.8 / 59.2 | 79.3 / 77.7 | 61.5 / 59.8 | 75.4 / 72.1 | 60.3 / 54.2 |
| Construction Project Management | 51.1 / 54.7 | 41.7 / 41.7 | 30.2 / 34.5 | 43.2 / 43.2 | 36.7 / 38.1 | 44.6 / 48.2 | 41.7 / 36.7 |
| Elementary Chinese | 53.2 / 58.7 | 29.4 / 34.9 | 28.5 / 28.5 | 57.9 / 61.1 | 45.6 / 44.8 | 48.0 / 44.4 | 44.8 / 42.1 |
| Elementary Commonsense | 68.2 / 73.7 | 46.5 / 49.5 | 35.6 / 45.5 | 62.6 / 71.2 | 52.5 / 49.0 | 55.6 / 56.1 | 50.5 / 48.0 |
| Ethnology | 63.7 / 74.1 | 42.2 / 46.7 | 36.3 / 39.3 | 65.9 / 59.3 | 48.1 / 42.2 | 63.0 / 55.6 | 47.4 / 45.2 |
| High School Politics | 67.1 / 65.7 | 44.1 / 49.0 | 35.7 / 41.3 | 76.9 / 67.8 | 49.0 / 50.3 | 53.8 / 51.7 | 49.0 / 53.8 |
| Modern Chinese | 56.0 / 62.1 | 34.5 / 40.5 | 28.4 / 30.2 | 45.7 / 45.7 | 44.0 / 39.7 | 41.4 / 45.7 | 40.5 / 38.8 |
| Traditional Chinese Medicine | 58.4 / 60.5 | 38.4 / 42.2 | 31.9 / 30.8 | 55.1 / 52.4 | 48.1 / 53.5 | 48.6 / 46.5 | 48.1 / 44.9 |
| Agronomy | 66.3 / 67.5 | 46.2 / 50.9 | 35.5 / 39.6 | 58.0 / 61.5 | 46.7 / 42.6 | 56.2 / 55.0 | 47.3 / 48.5 |
| Clinical Knowledge | 68.8 / 72.2 | 42.2 / 43.5 | 36.7 / 38.0 | 51.5 / 51.1 | 44.3 / 40.1 | 45.1 / 43.9 | 40.5 / 42.6 |
| College Medicine | 72.2 / 75.8 | 39.6 / 44.7 | 26.7 / 33.0 | 56.4 / 56.0 | 42.9 / 45.1 | 40.3 / 45.4 | 44.7 / 41.0 |
| Computer Security | 87.7 / 85.4 | 63.7 / 73.7 | 40.4 / 45.0 | 66.1 / 68.4 | 56.1 / 56.1 | 71.3 / 68.4 | 63.2 / 54.4 |
| Elementary IT | 93.7 / 94.5 | 76.9 / 77.7 | 54.6 / 63.3 | 79.0 / 75.6 | 68.1 / 63.9 | 73.5 / 74.8 | 66.0 / 63.0 |
| Food Science | 74.1 / 76.2 | 53.1 / 56.6 | 39.2 / 43.4 | 60.1 / 60.8 | 49.7 / 43.4 | 55.2 / 49.7 | 47.6 / 46.2 |
| Human Sexuality | 72.2 / 69.8 | 60.3 / 62.7 | 45.2 / 48.4 | 61.1 / 61.9 | 48.4 / 43.7 | 61.1 / 60.3 | 52.4 / 42.9 |
| Legal And Moral Basis | 91.1 / 91.1 | 82.7 / 85.5 | 67.3 / 73.8 | 92.1 / 93.0 | 83.6 / 82.2 | 90.2 / 90.2 | 84.6 / 77.1 |
| Nutrition | 73.8 / 72.4 | 49.7 / 56.6 | 42.1 / 42.8 | 57.9 / 64.8 | 53.1 / 47.6 | 52.4 / 54.5 | 51.0 / 43.4 |
| Professional Medicine | 66.5 / 67.3 | 34.8 / 37.2 | 26.6 / 32.7 | 50.5 / 50.5 | 37.5 / 36.7 | 41.0 / 39.6 | 33.0 / 34.0 |
| Sports Science | 70.9 / 72.1 | 51.5 / 57.0 | 43.6 / 43.0 | 60.0 / 60.0 | 49.7 / 49.1 | 60.6 / 63.0 | 50.3 / 47.9 |
| Business Ethics | 70.8 / 73.7 | 56.9 / 62.7 | 40.2 / 43.5 | 59.8 / 55.5 | 46.4 / 42.6 | 56.5 / 59.8 | 52.6 / 46.4 |
| College Education | 79.4 / 83.2 | 62.6 / 69.2 | 55.1 / 53.3 | 72.9 / 76.6 | 64.5 / 68.2 | 72.9 / 72.9 | 66.4 / 56.1 |
| Economics | 84.9 / 84.9 | 55.3 / 57.9 | 48.4 / 49.1 | 62.3 / 64.2 | 46.5 / 44.0 | 55.3 / 56.6 | 52.8 / 47.8 |
| Education | 63.8 / 64.4 | 51.5 / 53.4 | 41.7 / 44.2 | 69.9 / 70.6 | 60.1 / 60.7 | 60.1 / 61.3 | 58.9 / 57.7 |
| High School Geography | 78.0 / 75.4 | 42.4 / 51.7 | 44.1 / 42.4 | 66.1 / 67.8 | 47.5 / 54.2 | 56.8 / 55.1 | 47.5 / 52.5 |
| Journalism | 68.0 / 69.2 | 54.1 / 61.0 | 43.0 / 45.3 | 59.3 / 62.2 | 52.9 / 48.3 | 55.8 / 54.1 | 52.9 / 51.7 |
| Management | 82.9 / 84.3 | 56.7 / 64.8 | 49.5 / 49.5 | 68.6 / 71.9 | 62.9 / 61.0 | 65.2 / 67.6 | 62.4 / 59.0 |
| Marketing | 81.7 / 81.7 | 65.6 / 66.1 | 43.9 / 54.4 | 68.7 / 63.3 | 57.2 / 56.7 | 67.2 / 66.7 | 53.0 / 54.4 |
| Professional Accounting | 72.6 / 76.6 | 51.4 / 61.7 | 41.1 / 50.3 | 70.3 / 72.0 | 56.6 / 54.9 | 55.4 / 59.4 | 57.7 / 56.6 |
| Professional Psychology | 81.9 / 81.9 | 50.0 / 62.5 | 42.2 / 50.9 | 70.3 / 72.4 | 55.6 / 58.6 | 68.5 / 68.5 | 58.2 / 59.1 |
| Public Relations | 63.8 / 67.2 | 56.9 / 62.1 | 46.0 / 52.3 | 64.4 / 55.7 | 51.1 / 53.4 | 55.2 / 58.0 | 51.7 / 51.7 |
| Security Study | 80.0 / 80.7 | 54.8 / 67.4 | 48.1 / 48.9 | 70.4 / 73.3 | 58.5 / 63.7 | 64.4 / 62.2 | 60.7 / 62.2 |
| Sociology | 72.1 / 73.0 | 59.3 / 64.2 | 41.2 / 47.8 | 64.2 / 68.1 | 51.3 / 47.3 | 58.8 / 59.3 | 49.1 / 46.0 |
| Arts | 74.4 / 77.5 | 58.8 / 63.1 | 50.6 / 53.1 | 83.1 / 83.1 | 66.2 / 68.1 | 75.6 / 71.9 | 69.4 / 61.3 |
| College Law | 59.3 / 63.0 | 39.8 / 42.6 | 31.3 / 35.4 | 55.6 / 54.6 | 45.4 / 42.6 | 47.2 / 50.0 | 42.6 / 46.3 |
| Global Facts | 71.8 / 77.9 | 49.0 / 58.4 | 39.5 / 46.7 | 71.1 / 64.4 | 57.0 / 49.0 | 64.4 / 61.7 | 51.7 / 52.3 |
| International Law | 61.1 / 64.3 | 49.7 / 51.4 | 40.0 / 36.8 | 56.2 / 51.9 | 38.4 / 34.6 | 47.6 / 48.6 | 41.1 / 39.5 |
| Jurisprudence | 71.0 / 73.0 | 58.4 / 59.4 | 39.4 / 44.0 | 63.0 / 64.0 | 53.0 / 52.6 | 59.4 / 59.6 | 53.0 / 49.9 |
| Logical | 70.7 / 80.5 | 54.5 / 61.8 | 35.8 / 35.8 | 59.3 / 56.9 | 48.0 / 41.5 | 54.5 / 51.2 | 41.5 / 42.3 |
| Marxist Theory | 78.8 / 82.0 | 60.8 / 67.2 | 50.3 / 48.1 | 76.2 / 81.0 | 56.6 / 60.3 | 69.8 / 66.1 | 56.6 / 55.0 |
| Philosophy | 69.5 / 72.4 | 61.0 / 64.8 | 52.4 / 54.3 | 68.6 / 66.7 | 59.0 / 59.0 | 70.5 / 68.6 | 53.3 / 53.3 |
| Professional Law | 53.6 / 54.0 | 37.4 / 43.1 | 29.4 / 28.9 | 50.2 / 47.9 | 41.7 / 39.3 | 48.8 / 45.5 | 40.3 / 40.8 |
| World History | 84.5 / 83.9 | 64.0 / 65.8 | 45.3 / 49.1 | 64.6 / 68.9 | 55.3 / 57.8 | 76.4 / 75.2 | 56.5 / 58.4 |
| World Religions | 78.8 / 83.8 | 61.3 / 66.9 | 49.4 / 51.2 | 72.5 / 73.8 | 58.8 / 58.1 | 63.7 / 61.3 | 55.0 / 53.8 |
| Anatomy | 69.6 / 67.6 | 33.8 / 32.4 | 25.3 / 34.0 | 48.6 / 48.6 | 34.5 / 35.1 | 34.5 / 33.8 | 35.1 / 35.1 |
| Astronomy | 55.8 / 60.0 | 37.6 / 43.6 | 26.7 / 33.3 | 41.2 / 41.8 | 31.5 / 32.7 | 37.0 / 33.9 | 36.4 / 34.5 |
| College Actuarial Science | 43.4 / 41.5 | 28.3 / 32.1 | 32.1 / 28.3 | 30.2 / 30.2 | 23.6 / 23.6 | 27.4 / 30.2 | 25.5 / 31.1 |
| College Engineering Hydrology | 66.0 / 71.7 | 50.0 / 47.2 | 40.6 / 42.5 | 51.9 / 60.4 | 36.8 / 38.7 | 50.0 / 47.2 | 39.6 / 33.0 |
| College Mathematics | 45.7 / 45.7 | 23.8 / 30.5 | 24.8 / 27.6 | 24.8 / 26.7 | 21.9 / 29.5 | 36.2 / 31.4 | 28.6 / 27.6 |
| College Medical Statistics | 73.6 / 76.4 | 47.2 / 54.7 | 32.1 / 32.1 | 51.9 / 53.8 | 46.2 / 45.3 | 53.8 / 55.7 | 44.3 / 42.5 |
| Computer Science | 77.9 / 82.4 | 52.9 / 58.3 | 34.3 / 42.6 | 58.3 / 58.8 | 47.1 / 48.0 | 55.9 / 53.9 | 48.0 / 46.6 |
| Conceptual Physics | 73.5 / 74.1 | 47.6 / 54.4 | 38.8 / 38.1 | 60.5 / 57.1 | 63.3 / 64.6 | 51.0 / 48.3 | 63.9 / 58.5 |
| Electrical Engineering | 65.1 / 70.3 | 47.1 / 53.5 | 40.1 / 37.2 | 54.1 / 55.2 | 37.8 / 41.3 | 55.2 / 54.7 | 45.9 / 43.6 |
| Elementary Mathematics | 51.7 / 51.7 | 33.5 / 31.3 | 28.3 / 27.0 | 41.3 / 40.0 | 45.7 / 35.2 | 28.7 / 27.0 | 40.4 / 40.9 |
| Genetics | 68.8 / 71.6 | 45.5 / 54.5 | 32.4 / 38.1 | 46.0 / 49.4 | 40.3 / 41.5 | 44.9 / 44.9 | 41.5 / 40.3 |
| High School Biology | 64.5 / 66.9 | 38.5 / 43.8 | 26.0 / 30.8 | 59.2 / 56.8 | 60.9 / 63.9 | 52.1 / 48.5 | 62.7 / 58.0 |
| High School Chemistry | 44.7 / 53.0 | 25.0 / 31.1 | 28.0 / 29.5 | 44.7 / 40.9 | 55.3 / 58.3 | 34.8 / 36.4 | 52.3 / 48.5 |
| High School Mathematics | 45.7 / 48.8 | 28.0 / 29.3 | 21.3 / 27.4 | 25.6 / 33.5 | 34.8 / 28.7 | 34.8 / 28.0 | 35.4 / 31.1 |
| High School Physics | 70.0 / 68.2 | 38.2 / 42.7 | 28.2 / 30.0 | 41.8 / 40.9 | 47.3 / 44.5 | 37.3 / 40.9 | 45.5 / 46.4 |
| Machine Learning | 77.9 / 80.3 | 48.4 / 50.0 | 31.1 / 32.0 | 51.6 / 48.4 | 45.1 / 41.0 | 54.1 / 57.4 | 41.0 / 41.8 |
| Virology | 79.3 / 78.7 | 58.6 / 60.4 | 34.9 / 42.0 | 63.3 / 63.9 | 49.1 / 50.3 | 55.0 / 53.8 | 47.3 / 49.1 |

Table 11: Zero-shot accuracy of models. We report macro average accuracy over subjects within each category. "Overall" = macro average score over all subjects. "State" indicates whether the model is pre-trained (Base) or Fine-tuned to follow instructions (Chat). '*' indicate there are both Base and Chat model released, we choose the one with better overall accuracy. The first block is multilingual- or English-oriented models, and the second block is Chinese-oriented models. To save space, we didn't present models with an overall score lower than 30.

| Model | State | STEM | Humanities | Social Science | Other | China-specific | Overall |
|-------|-------|------|------------|----------------|-------|----------------|---------|
| GPT4 | Chat | 63.13 | 69.19 | 70.26 | 73.16 | 63.47 | 68.89 |
| ChatGPT | Chat | 44.80 | 53.61 | 54.22 | 59.95 | 49.74 | 53.22 |
| LLaMA2-70B* | Base | 40.23 | 53.41 | 50.10 | 52.91 | 45.16 | 48.87 |
| BLOOMZ-7B | Chat | 33.03 | 45.74 | 45.74 | 46.25 | 41.58 | 42.80 |
| Falcon-40B | Base | 31.11 | 41.30 | 40.87 | 40.61 | 36.05 | 38.50 |
| LLaMA2-13B* | Chat | 31.57 | 37.89 | 38.10 | 39.00 | 35.44 | 36.60 |
| LLaMA-65B | Base | 31.09 | 34.45 | 36.05 | 37.94 | 32.89 | 34.88 |
| BX$_{\text{LLaMA}}$-30B | Chat | 28.79 | 32.61 | 31.65 | 34.22 | 31.47 | 31.69 |
| LLaMA-30B | Base | 30.02 | 31.87 | 31.51 | 32.90 | 29.64 | 31.54 |
| BX$_{\text{LLaMA}}$-13B | Chat | 26.46 | 29.36 | 31.81 | 31.55 | 29.17 | 30.06 |
| Baichuan2-13B* | Base | 47.59 | 65.57 | 65.24 | 65.47 | 62.10 | 60.88 |
| Xverse-13B* | Base | 43.42 | 60.51 | 60.65 | 64.20 | 56.69 | 57.04 |
| InternLM-20B* | Chat | 43.68 | 61.78 | 58.19 | 57.54 | 55.26 | 55.06 |
| Baichuan-13B* | Base | 41.63 | 60.26 | 59.62 | 56.15 | 56.03 | 54.40 |
| InternLM-7B* | Base | 43.04 | 56.72 | 56.96 | 54.50 | 54.55 | 52.83 |
| ChatGLM2-6B | Chat | 42.98 | 52.42 | 52.56 | 52.15 | 49.38 | 50.01 |
| BatGPT-15B | Chat | 43.15 | 50.91 | 52.66 | 52.23 | 49.09 | 49.81 |
| Baichuan-7B | Base | 32.79 | 44.43 | 46.83 | 44.79 | 43.19 | 42.35 |
| ChatGLM-6B | Chat | 32.54 | 42.91 | 44.91 | 42.29 | 42.08 | 40.80 |
| Random | – | 25.00 | 25.00 | 25.00 | 25.00 | 25.00 | 25.00 |

Table 12: The Impact of Chain of Thoughts (COT) on the performance of several LLMs on CMMLU. The numbers on the left represent the values after incorporating COT, with the values in parentheses indicating the change relative to the model's performance in the 0-shot scenario.

| Model | STEM | Humanities | Social Science | Other | China-specific | Overall |
|-------|------|------------|----------------|-------|----------------|---------|
| Baichuan2-13B-Chat | 42.7 (-2.5) | 57.7 (-6.3) | 56.0 (-8.0) | 55.4 (-6.6) | 53.8 (-7.7) | 52.8 (-6.0) |
| BatGPT-15B-sirius | 34.7 (-3.5) | 44.2 (-2.6) | 45.8 (-2.2) | 46.6 (-1.2) | 43.6 (-1.3) | 42.9 (-2.4) |
| ChatGLM-6B | 29.9 (-2.3) | 37.9 (-4.8) | 39.6 (-4.6) | 36.2 (-6.1) | 38.3 (-3.4) | 36.0 (-4.4) |
| ChatGLM2-6B | 42.6 (+0.1) | 52.3 (+0.3) | 51.3 (-0.9) | 51.6 (-0.3) | 49.0 (+0.2) | 49.3 (-0.3) |
| ChatGPT | 46.6 (+1.4) | 52.5 (-1.0) | 54.0 (-0.3) | 58.0 (-2.0) | 47.7 (-2.2) | 52.7 (-0.4) |
| InternLM-Chat-20B | 32.3 (-9.8) | 48.1 (-10.7) | 48.1 (-9.8) | 44.6 (-11.0) | 44.9 (-9.4) | 43.3 (-10.2) |
| Xverse-13B-Chat | 30.5 (-9.6) | 40.2 (-16.1) | 43.0 (-14.3) | 42.8 (-15.3) | 38.7 (-14.3) | 39.3 (-13.7) |

