# OpenReview forum: "CMMLU: Measuring massive multitask language understanding in Chinese"
_ICLR.cc/2024/Conference — Submitted to ICLR 2024_

### Official Review · Reviewer_ZnNq · 2023-10-28

**Soundness:** 3 good
**Presentation:** 3 good
**Contribution:** 3 good
**Rating:** 5
**Confidence:** 4

**Summary:**

This work proposes a  Chinese multi-task benchmark dataset CMMLU  for better evaluating the language understanding ability of LLMs in the context of Chinese. Compared to previous benchmark datasets, besides the general tasks, CMMLU consists of many Chinese-specific tasks.

Meanwhile, this work has also conducted a lot of experiments to check the performance of the 20 most popular non-Chinese-specific and Chinese-specific LLMs.  The experimental results provide a good reference for developers to choose the LLM in the context of Chinese.

**Strengths:**

1. CMMLU is specifically designed for evaluating Chinese LLMs. It not only consists of general natural language understanding tasks, but also some region-specific tasks such as Chinese driving rules, food culture, and qualifications. Thus, CMMLU can better reveal the real LLM performance in the Chinese scenarios.

2. This work invested many efforts in collecting non-publicly available questions to reduce the possibility that the collected questions have already been learned by LLMs.

3. This work has evaluated many multilingual LLMs and many Chinese LLMs at the same time. Meanwhile, the authors also compare the best Chinese LLM Baichuan2-13B with the best LLM GPT4 by subjects. This comparison can answer the question of why we need Chinese LLMs/benchmarks in the Chinese scenarios.

4. Many deep analyses have shown many interesting and useful findings.

**Weaknesses:**

1. Although this work is technically sound and solid, CMMLU lacks enough novelty or other special contributions.  The major highlight is that CMMLU consists of some Chinese-specific tasks. This is more or less like an A+B incremental work.

2. All questions are formatted as multiple-choice with 4 choices.   This may make it difficult to comprehensively test the performance of LLMs.

3. The experimental methodology of most experiments is language-agnostic.  It only simply compares general LLMs vs Chinese LLMs and non-Chinese-specific tasks vs  Chinese-specific tasks.  I think more experiments should be deeply combined with the Chinese cultural and Chinese linguistic characteristics.

4. This work needs to analyze the correlation between the performance reported by CMMLU and the real performance measured in the representative downstream NLU tasks. Otherwise, it is difficult to determine whether CMMLU can reflect the NLU performance of a LLM.

**Questions:**

Besides the questions in Weakness, there are some minor questions:

1) What form will this dataset be released in the future?

2) Besides the Chinese-specific tasks and data source, is there any other Chinese-specific feature that has been considered in CMMLU?

**Details Of Ethics Concerns:**

This article involves data collection, especially non-public data collection.  This faces some copyright risks.

However, I did not see how to address these ethics issues in this work.

---

> ### Author Response · Authors · 2023-11-17
> **Thank you for your comments**
>
> Thank you for your insightful feedback. We appreciate the opportunity to clarify and expand upon certain aspects of our work.
>
> > Novelty and Special Contributions
>
> While we acknowledge that the paper is inspired by the MMLU paper, CMMLU is distinct in its data and analysis.  Our dataset was independently developed, not translated from existing English datasets. The analyses we conducted differ significantly from the MMLU paper and are specifically tailored to current LLMs. We believe our analysis offers unique insights and contribute meaningfully to LLM research.
>
>
> > All questions are formatted as multiple-choice with 4 choices. This may make it difficult to comprehensively test the performance of LLMs.
>
> We acknowledge that the multiple-choice format of testing has its limitations. However, this format offers a standardized and accessible evaluation method, as evidenced by its prevalence in popular LLM evaluation frameworks and leaderboards, such as the one hosted by Hugging Face (https://huggingface.co/spaces/HuggingFaceH4/open_llm_leaderboard), of which around half is  multiple-choice tasks. Additionally, multiple-choice questions can be converted to a perplexity comparison of the different choices, which is the dominant approach for pre-trained LLMs evaluation, given that this is how instruction fine-tuning is carried out. The decision to use a multiple-choice format was made after careful consideration.
>
> Currently, there is no universally-recognized benchmark capable of comprehensively evaluating the performance of LLMs. Most existing leaderboards use a combination of multiple datasets to provide a holistic assessment of LLMs. In this context, CMMLU is not intended to provide a holistic evaluation of LLMs, but rather be one of several benchmarks in a comprehensive evaluation strategy.
>
>
> > The experimental methodology of most experiments is language-agnostic. It only simply compares general LLMs vs Chinese LLMs and non-Chinese-specific tasks vs Chinese-specific tasks. I think more experiments should be deeply combined with the Chinese cultural and Chinese linguistic characteristics.
>
> Your suggestion to incorporate a more in-depth exploration of Chinese cultural and linguistic characteristics is well-received. To avoid misunderstanding, we want to first clarify that all data in CMMLU is in Mandarin Chinese. Our initial focus was to establish a broad comparison between multilingual and Chinese-specific LLMs using CMMLU. This approach was intended to set a baseline for future, more detailed studies. We agree that delving deeper into the nuances of the Chinese language and culture will further enrich our understanding of LLM capabilities, but this is beyond the scope of this work.
>
>
> > This work needs to analyze the correlation between the performance reported by CMMLU and the real performance measured in the representative downstream NLU tasks. Otherwise, it is difficult to determine whether CMMLU can reflect the NLU performance of a LLM.
>
> We have included an analysis of the correlation between CMMLU performance and five prominent downstream English benchmarks in Appendix Section I. Our findings indicate that performance on CMMLU has a strong correlation with results on four of these benchmarks (with Pearson’s Correlation r > 0.94), which span areas such as mathematics, commonsense reasoning, and coding. The slight exception is the PIQA task (with Pearson’s Correlation still high at r = 0.88), where the task relevance is somewhat reduced due to most models achieving high scores (>80%) on this task.
>
>
> > What form will this dataset be released in the future?
>
> The CMMLU dataset will be released under a 'Creative Commons Attribution-NonCommercial-ShareAlike 4.0 International License' to facilitate open-source research and ensure accessibility for non-commercial use.
>
>
> > Besides the Chinese-specific tasks and data source, is there any other Chinese-specific feature that has been considered in CMMLU?
>
> No. Our focus is on the dataset tasks and sources, which we believe adequately address the knowledge-based evaluation needs of Chinese LLMs, consistent with MMLU for English and IndoMMLU for languages of Indonesia. We are open to exploring additional features in future iterations of CMMLU based on community feedback and evolving research needs.

---

> > ### Comment · Reviewer_ZnNq · 2023-11-19
> >
> > Thanks for your comments.
> >
> > Your answers have addressed some mentioned issues and I have raised the contribution from 2 to 3. Nonetheless, I still keep my overall recommendation because of the lack of enough novelty and contribution.
> >
> > Besides, after the initial review, I saw that CMMLU had been used in some PR scenarios, for example, BlueLM by vivo.   I don't know if this violates the anonymity policy.

---

> > > ### Author Response · Authors · 2023-11-20
> > >
> > > Thank you for your feedback and for raising our contribution score. We appreciate your recognition of the improvements we've made.
> > >
> > > Regarding the novelty and contribution of our work, we believe it offers significant advancements in the field considering two reasons. First, our benchmark has been widely adopted by numerous language model developers, which is a testament to its utility and relevance in the community. This widespread adoption underscores the originality and practical impact of our research. Second, CMMLU plays a crucial role in advancing inclusive language technology systems. Existing language model evaluation predominantly centers around English, but CMMLU strives to democratize these current research progress, thereby extending their benefits more broadly to the community.
> > >
> > > Concerning the anonymity policy, we have taken careful measures to comply with it. Firstly, we have not included any direct references or links to our preprint or GitHub in our paper or rebuttal. Secondly, we didn't mention our submission on social media, or online websites. We have no direct connection with BlueLM by vivo or any other organization that has used the datasets to benchmark their models and mentioned it in their press releases, and have had no role in any such publicity for the dataset. The anonymity policy is intended to prevent authors from revealing their identity, not to restrict others from using or mentioning it in the public domain. Thus, our conduct fully aligns with the requirements of the policy.
> > >
> > > We hope this addresses your concerns. Thank you.

---

### Official Review · Reviewer_qmrE · 2023-10-30

**Soundness:** 3 good
**Presentation:** 3 good
**Contribution:** 3 good
**Rating:** 6
**Confidence:** 4

**Summary:**

This paper introduced a fully Sinicized Chinese test benchmark, CMMLU, specifically designed to evaluate the knowledge and reasoning capabilities of language models in a Chinese context. CMMLU covered 67 topics ranging from basic disciplines to advanced professional levels, with answers specific to the Chinese region.

**Strengths:**

The paper conducted extensive experiments, including on the proprietary GPT-4 (even though OpenAI consistently updated GPT versions without much fanfare).

The content was detailed and held significant practical value for the Chinese domain.

**Weaknesses:**

However, an LLM passing professional exams doesn't necessarily indicate its true capabilities, raising concerns about construct validity.

The crisis of research replication based on language models was severe, and the evaluation methods had limitations.

Assessing the political biases inherent in the language models presented in the benchmark was challenging and required naturalistic observation.

**Questions:**

1. One concern I had was that this Chinese test benchmark did not include evaluation criteria for Chinese machine translation. Many studies are now focusing on evaluating the generalized machine translation capabilities of LLMs. Given the extensive work the authors did on this benchmark, how did authors view the evaluation criteria for Chinese translations?

2. The outputs of LLMs were uncertain. Even a minor change in a prompt could lead to variations in the output. In light of this benchmarking paper, how did the authors perceive this issue? How should the benchmark address the inherent unpredictability of LLMs?

3. Typically, the Chain-of-Thought method had proven successful on LLMs. However, this paper concluded that the Chain-of-Thought was not effective in enhancing model performance, which contradicted the feedback received from practical use of LLMs with the Chain-of-Thought. A more detailed analysis and explanation were requested.

4. LLMs demonstrated strong In-Context Learning capabilities. It would be worth exploring whether adding appropriate knowledge to the prompt could answer benchmark questions to validate the benchmark's effectiveness.

5. It was known that LLMs would respond cautiously to safety questions when posed in English. However, when asked in less common languages, they might provide bolder answers, potentially bypassing restrictions. Did the CMMLU safety benchmark consider addressing this phenomenon?

6. How did the authors ensure that the proposed test benchmark was free from data contamination?

---

> ### Author Response · Authors · 2023-11-17
> **Thank you for your comments**
>
> Thank you for your detailed review of our paper. We appreciate your concerns and have addressed each of them below with the aim of clarifying our approach and methodology.
>
> > An LLM passing professional exams doesn't necessarily indicate its true capabilities, raising concerns about construct validity.
>
> To comprehensively evaluate the capabilities of an LLM, evaluation needs to encompass aspects including knowledge, commonsense reasoning, fairness, and generative capabilities. This paper focuses on evaluating knowledge-related aspects in the Chinese language, as a component of a broader evaluation, consistent with the use of MMLU in English.
>
> School exam questions have been extensively employed as a proxy for evaluating knowledge in LLMs. Examples include the use of the English MMLU dataset in the official release documentation/technical reports for GPT-4, LLaMA-2, Falcon, and various other LLMs, and the HuggingFace Open LLM leaderboard (https://huggingface.co/spaces/HuggingFaceH4/open_llm_leaderboard), the most popular LLM leaderboard today.
>
> We believe that developing such a benchmark for Mandarin Chinese is essential for more inclusive evaluation of LLMs beyond the English language.
>
>
> > The crisis of research replication based on language models was severe, and the evaluation methods had limitations.
>
> We absolutely share your concerns regarding the importance of reproducibility in research, but do not agree that this is a limitation of the dataset. To ensure transparency, we have meticulously documented the experimental settings in Section 4, including the number of shots for few-shot learning, and the exact prompt used. As such, all experiment results in the paper can be reproduced. Additionally, a number of subsequent studies have successfully replicated our results based on the arXiv preprint version of this paper, and we've also made our code completely public. In adherence with the policy of anonymity, we refrain from specifically referencing these papers here as they cite the arXiv preprint, compromising the anonymity of this paper, but we would be happy to provide the links after the review period.
>
> Additionally, the issue of LLM research replication is an issue with models and model training rather than evaluation datasets, which is the focus of this paper.
>
> > Assessing the political biases inherent in the language models presented in the benchmark was challenging and required naturalistic observation.
>
> Our paper focuses on knowledge evaluation, not assessing the political biases of LLMs, which is an orthogonal dimension of LLM evaluation, as stated above.
>
>
> > One concern I had was that this Chinese test benchmark did not include evaluation criteria for Chinese machine translation.
>
> Once again, this benchmark specifically focuses on knowledge evaluation, as a critical component of comprehensive LLM evaluation. This focus is consistent with earlier work such as English MMLU (https://arxiv.org/pdf/2009.03300.pdf) at ICLR 2021 and Indonesian MMLU (https://arxiv.org/pdf/2310.04928.pdf)  at EMNLP 2023.  Machine translation is an instance of a task which could be included in a test suite to separately evaluate the generative abilities of the LLM, as mentioned above. That is, we agree that machine translation is an important aspect of broader LLM evaluation, but it is orthogonal to knowledge-based evaluation.
>
>
> > Inherent unpredictability of LLMs and prompt sensitive
>
> This is a more general issue with any evaluation of LLMs, and not specific to our benchmark. We have strived to mitigate this by employing simple and commonly-used prompts, to minimize variance in model outputs. This approach leads to fairer comparison of different models. Also of note is that we tested several (i.e., 3) widely-used prompts for multiple choice questions on more than 5 models. We find that the standard deviation in the average score is <1%.
>
>
> > Typically, the Chain-of-Thought method has proven successful on LLMs. However, this paper concluded that the Chain-of-Thought was not effective in enhancing model performance, which contradicted the feedback received from the practical use of LLMs with the Chain-of-Thought. A more detailed analysis and explanation were requested.
>
> The effectiveness of Chain-of-Thought (COT) in LLMs primarily comes about in tasks requiring reasoning. For memory-intensive tasks (e.g., recalling historical facts), COT can not only be less effective but potentially impede performance.
>
> In the CMMLU benchmark, the majority of tasks are knowledge-intensive rather than reasoning-based. This explains why COT does not enhance performance in our case. Furthermore, as discussed in Section 4.2 of our paper, issues like regex matching failures can further diminish effectiveness.

---

> > ### Author Response · Authors · 2023-11-17
> > **For last few questions**
> >
> > > LLMs demonstrated strong In-Context Learning capabilities. It would be worth exploring whether adding appropriate knowledge to the prompt could answer benchmark questions to validate the benchmark's effectiveness.
> >
> > In-context learning is indeed a vital aspect of LLM evaluation, often utilized to convey task requirements to the model. In our benchmark, which focuses on knowledge-intensive tasks, in-context learning primarily serves to clarify task objectives rather than contribute additional instruction on how to answer the question. This is also the standard approach to in-context learning in evaluation work, i.e. to provide a few example questions and answers before asking the question. In this paper, we performed substantial analysis of this kind of in-context learning (Section 4).
> >
> > Your suggestion to explore the addition of knowledge to prompts to validate the benchmark's effectiveness is valuable. Our focus is on assessing the inherent knowledge of LLMs, as acquired during pre-training and fine-tuning. Incorporating external knowledge would not align with this objective, as it would not accurately reflect the model's intrinsic capabilities. However, we acknowledge that exploring in-context learning (with external knowledge) could be a direction for future research. Reading comprehension (RC) tasks are well represented in existing RC datasets and could be used for this.
> >
> >
> > > Did the CMMLU safety benchmark consider addressing this phenomenon?
> >
> > Once again, CMMLU is focused on knowledge evaluation, consistent with MMLU and IndoMMLU. Safety considerations are a vital consideration in the deployment of any LLM, but not part of knowledge evaluation.
> >
> >
> > > How did the authors ensure that the proposed test benchmark was free from data contamination?
> >
> > As mentioned in Section 3 “Data Collection”, more than 80% of our data was crawled from PDFs (after OCR), reducing the possibility of it occurring in LLM training data. Indeed, our specific efforts to mitigate the effects of data contamination are acknowledged by Reviewer ZnNq (See the strengths part). Moreover, the low performance for most subjects suggests that there is little data contamination in LLM pre-training.
> >
> > We hope these responses adequately address your concerns. We are committed to improving our work based on your valuable feedback.

---

> ### Comment · Reviewer_qmrE · 2023-11-20
>
> The authors adequately explained the issues raised and all my other concerns were answered in the comments and responses of the other reviewers. CMMLU has been used as a benchmark in many papers. I hope the authors will keep the dataset up to date in the future.
>
> Soundness changed to 3. Rating raised to 6.

---

> > ### Author Response · Authors · 2023-11-20
> >
> > Thanks for your advice and the acknowledgement of the value of this paper! We will keep updating the dataset.

---

### Official Review · Reviewer_CUim · 2023-11-11

**Soundness:** 3 good
**Presentation:** 3 good
**Contribution:** 4 excellent
**Rating:** 8
**Confidence:** 4

**Summary:**

This paper introduced CMMLU, a benchmark designed to assess the multi-task language understanding capabilities in Chinese. The authors ran the benchmark on various open-source and API-based models and performed extensive analysis to identify several factors that impact model performance and propose actionable directions for enhancing LLMs.

**Strengths:**

1. The CMMLU benchmark is very comprehensive, covering a wide range of subjects.

2. The paper addresses the significant gap in evaluating Chinese language and cultural context understanding, a critical aspect given the dominance of English-centric benchmarks.

3. The work can be very useful for Chinese LLM community.

4. The paper provides an in-depth analysis of the performance of various LLMs, under different evaluation settings.

5. The paper also provides very interesting findings in terms of chain-of-thought, SFT/RLHF, etc.

**Weaknesses:**

1. A human baseline is lacking for the benchmark. It'd be great to see what level of accuracy human can get on the benchmark.

2. There's no discussion on the difficulty distribution of questions in each subset. A well designed benchmark or test should cover questions spanning all difficulty levels from the easiest to the hardest. It's unknown what the difficulty distribution is for each subset. If difficulty distribution is very centric (for example, all samples in a subset are all very easy or very hard), then models will be likely to get them all correct or all wrong, which cannot provide a **smooth** estimation of the model's ability. A non-smooth evaluation can also be related to the phenomenon of "emergent ability". See me question 2.

**Questions:**

1. In page 1, "numerous tasks within CMMLU have answers speciﬁc to China, which may not be universally applicable or considered correct in other regions or languages.". Do you think it would be a good idea to have questions **with answers that are not generally agreed upon worldwide** in the datasets? How many samples of this kind are there in the benchmark?

2. When you evaluate open-source models, have you seen "emergent ability" in terms of model's size? More precisely, are there some tasks that can only be solved by a large model? If so, then what are the difficulty distributions of those tasks?

---

> ### Author Response · Authors · 2023-11-17
> **Thank you for your comments**
>
> Thank you for your insightful and detailed review of our paper. We have tried to address your concerns by doing additional human annotation on answer agreement, and include 2 more sections relating to difficulty and emergent abilities, as detailed below.
>
> > human baseline
>
> Thank you for your suggestion on incorporating a human baseline. Given the wide range of knowledge areas covered in this benchmark, including STEM and social science, establishing a comprehensive human baseline is challenging. The broad knowledge spectrum makes it difficult for any single individual to provide a meaningful baseline. Moreover, recruiting subject matter experts for each domain, given the extensive scope of the benchmark, is not feasible.
>
> However, we appreciate the need to provide some form of human comparison. To address this, we propose using statistical data from exams as a proxy for human performance. In China, most school exams require a minimum passing score of 60%, drive licence exams require a score of 90%, etc., suggesting that a competent human test-taker would likely achieve at least that accuracy rate in each subject area. We can supplement this with additional statistics on average scores for various levels of tests to provide a more nuanced view of human performance, although this will be at an aggregate level from educational authorities
>
> We believe this approach, while not perfect, offers a practical and informative way to include a human baseline in our paper. It would provide readers with a point of reference to better understand the performance of LLMs in relation to human capabilities.
>
> We are open to further suggestions and would be grateful for your opinion on this proposed method.
>
>
> > Difficulty distribution of questions in each subset
>
> We appreciate your insight into the importance of question difficulty distribution. We've analyzed this from two perspectives. Firstly, the CMMLU benchmark encompasses a diverse range of difficulty levels: 5 subjects at primary school level, 10 at middle/high school level, 23 at college level, and 29 at professional level, ensuring a comprehensive difficulty spectrum.
>
> Secondly, to estimate the difficulty distribution within each subject, we evaluated the top 20 models from our main results table. Each question was treated as a data point, and we recorded the number of models that correctly answer each question. This approach allowed us to map out the difficulty distribution across subjects. We've added a section in the Appendix (Section C) to analyze this. Briefly, our findings show that most subjects exhibit a single peak in difficulty, indicating a smooth challenge gradient within those subjects. However, some areas, like machine learning and professional law, display bimodal difficulty distributions, suggesting a mix of easy and hard questions, with fewer intermediate ones. Additionally, the maximum for the distribution varies quite a bit, suggesting some subjects such as “arts” are generally easy/LLMs cover the domain knowledge well, while others such as “college mathematics” are much harder/LLMs do not have good coverage of the necessary domain knowledge.
>
>
> > Should we add answers that are not generally agreed upon worldwide to this dataset?
>
> Our stance is that a language-specific benchmark should reflect the knowledge and perspectives relevant to its target audience. For contentious historical or political topics, aligning with regional consensus is in line with the benchmark's purpose.
>
> We first analyzed the different subject areas, and concluded that such questions will only appear in education, history, politics related subjects..
>
> Through careful selection, we identified 7 subjects which may contain such questions:: Chinese history, world history, high school politics, world religions, journalism, legal and moral basis, and Marxist theory. The total number of questions across these subjects is 1362. We then sampled 10%=136 questions to do human annotation, sourcing an expert with a Master's degree in International Relations, currently researching Chinese history, world history, and politics. Their role was to identify questions whose answers might not be universally accepted or agreed upon globally.
>
> This annotation process revealed that 19 questions within our sample had answers potentially subject to regional or cultural differences in interpretation. Based on this finding, we estimate that approximately 190 questions in our dataset (roughly 2% in total across the 7 subjects) might have answers that are not universally agreed upon. Given this small proportion, we believe that the presence of such questions will not significantly impact the overall evaluation results of our study
>
> We can add this analysis to the paper if you consider it will enhance the content.

---

> > ### Author Response · Authors · 2023-11-17
> > **For your last question**
> >
> > > "emergent ability" in terms of model's size.
> >
> > Recent studies such as "Are Emergent Abilities in Large Language Models just In-Context Learning?" (https://arxiv.org/abs/2309.01809) and “Are Emergent Abilities of Large Language Models a Mirage?” (https://arxiv.org/abs/2304.15004) have raised critical concerns about “emergent abilities”, suggesting that what had previously been identified as emergent abilities in large language models may merely have been the consequence of discretization effects in evaluation metrics, in-context learning, or the effects of memorability, data leakage, and quality. In our analysis, we have observed findings that align with this perspective.
> >
> > We assessed the concept of 'emergent ability' using the LLaMA-2 model family, and added a section to the Appendix (Section D) to discuss this. Our analysis indicates that while larger models generally show improved performance, it's both overall smooth, but also not consistent across all subjects. In fact, it’s only in a small minority of subjects like college education and human sexuality, that performance jumps between smaller (7B, 13B) and larger (70B) models. While this study is limited to LLaMA-2 and there is further analysis that could be done, these results are not supportive of the hypothesis of emergent abilities.
> >
> >
> > We are very appreciate of your insightful reviews, and hope these new analyses answer your questions.

---

> > > ### Comment · Reviewer_CUim · 2023-11-19
> > >
> > > Thank you for your reply.
> > >
> > > **human baseline**
> > >
> > > Regaring the human baselines, I don't think the commonly used passing scores in real life (e.g, 60% for school exams, and 90% for driving licence exams) are good indicators of human average scores. Those passing scores are usually the minimum standard or a requirement for practiced learners, which cannot represent the **averaged** ability of a group of people.
> > >
> > > But just as you said, setting the human baseline can be challenging. I think it'd be great if the paper can provide such baselines, but it's still OK if those are not available.
> > >
> > > **Other questions**
> > >
> > > You reply has addressed my other concerns.
> > >
> > > **Score Change**
> > >
> > > Given you additional results and reply, I changed my scores as follows:
> > >
> > > Contribution: 3 --> 4
> > >
> > > Rating: 6 --> 8

---

> > > > ### Author Response · Authors · 2023-11-20
> > > >
> > > > Thank you very much for the questions you raised. The additional content added to answer your questions has greatly enhanced the paper.

---

### Meta-Review · Area_Chair_aSES · 2023-12-06

**Metareview:**

This paper presents CMMLU, a new Mandarin Chinese benchmark for assessing the capabilities of LLMs that covers various subjects, including natural sciences, social sciences, engineering, and the humanities. The authors present a detailed evaluation of a large number of LLMs and perform extensive analyses of model performance. The dataset is novel and the work is technically sound and solid. It is a nice extension to previous work on the English MMLU and Indonesian MMLU.

The main issue here, however, is the fact that they authors have revealed their identities during the rebuttal. Specfically, they include more than one mentions to their arxiv paper; e.g.:
“Additionally, a number of subsequent studies have successfully replicated our results based on the arXiv preprint version of this paper, and we've also made our code completely public. In adherence with the policy of anonymity, we refrain from specifically referencing these papers here as they cite the arXiv preprint, compromising the anonymity of this paper, but we would be happy to provide the links after the review period.”

The authors seem to have missed what the anonymity policy entails. While they do not directly reveal their names, they do point out their arxiv submission in their response. In ICLR’s 2024 author guide, there is a question specific to this issue, and it reads as follows (https://iclr.cc/Conferences/2024/AuthorGuide):
“Q. I have a nearly identical version on arxiv.  Does this violate the anonymity policy?
No, so long as you do not refer to it explicity.”

The authors have directly refered to their arxiv submission in the rebuttal, and, therefore, they have violated the anonymity policy of ICLR 2024.
The extent to which this has affected the reviewers’ decision is not clear. I do, however, note that 2/3 reviewers have commented on the arxiv paper after having read the author response, with one of the pointing out the anonymity issue.
Furthermore, I would not think that the fact that “CMMLU has been used as a benchmark in many papers” is an argument that should be made in favour of a paper as part of the peer-review process, and is certainly not a scientific argument.

Additional comments:
The authors have not addressed the last reviewer’s ethics concerns (e.g. are there copyright issues?).

An issue is raised with respect to the following: “questions within our sample had answers potentially subject to regional or cultural differences in interpretation”. The authors have added a very small scale study to assess this but the extent to which this is actually representative of the distribution in the full dataset is less clear. How was this 10% sampled exactly? This is a crucial bit of information that is missing.

Regarding the human baseline, I’d think that this refers to experts (and not e.g., test takers), e.g., what is the inter-annotator agreement on a certain exam test set (which can be considered an upper-bound for machine performance).

In the rebuttal, in response to one of the reviewer questions, the authors mention that “In the CMMLU benchmark, the majority of tasks are knowledge-intensive rather than reasoning-based”. However, in the very abstract of the paper, the authors metion that “CMMLU fills the gap in evaluating the knowledge and reasoning capabilities of large language models in the Chinese context”. It therefore sounds like this is a misleading claim. The authors need to clarify further.

**Justification For Why Not Higher Score:**

There seems to be an anonymity issue with this work.

**Justification For Why Not Lower Score:**

NA

---

### Decision · Program_Chairs · 2024-01-16

Reject